# Myosin II mediates Shh signals to shape dental epithelia via control of cell adhesion and movement

Wei Du[1,2☉], Adya Verma[3☉], Qianlin Ye[2], Wen Du[1], Sandy Lin[2], Atsushi Yamanaka[4], Ophir D. Klein[3,5], Jimmy K. Hu[2,6]*

**1** State Key Laboratory of Oral Diseases & National Center for Stomatology & National Clinical Research Center for Oral Diseases, West China Hospital of Stomatology, Sichuan University, Chengdu, Sichuan, China, **2** School of Dentistry, University of California Los Angeles, Los Angeles, California, United States of America, **3** Department of Orofacial Sciences, University of California San Francisco, San Francisco, California, United States of America, **4** Department of Oral Anatomy and Cell Biology, Graduate School of Medical and Dental Sciences, Kagoshima University, Kagoshima, Japan, **5** Department of Pediatrics, Cedars-Sinai Medical Center, Los Angeles, California, United States of America, **6** Molecular Biology Institute, University of California Los Angeles, Los Angeles, California, United States of America

☉ These authors contributed equally to this work.

* jkhu@ucla.edu

**Data Availability Statement:** All relevant data are within the manuscript and its Supporting Information files.

## Abstract

The development of ectodermal organs begins with the formation of a stratified epithelial placode that progressively invaginates into the underlying mesenchyme as the organ takes its shape. Signaling by secreted molecules is critical for epithelial morphogenesis, but how that information leads to cell rearrangement and tissue shape changes remains an open question. Using the mouse dentition as a model, we first establish that non-muscle myosin II is essential for dental epithelial invagination and show that it functions by promoting cell-cell adhesion and persistent convergent cell movements in the suprabasal layer. Shh signaling controls these processes by inducing myosin II activation via AKT. Pharmacological induction of AKT and myosin II can also rescue defects caused by the inhibition of Shh. Together, our results support a model in which the Shh signal is transmitted through myosin II to power effective cellular rearrangement for proper dental epithelial invagination.

## Author summary

During embryonic development, teeth, hair follicles, and glands are examples of ectodermally-derived organs that are formed from slightly thickened epithelial layers, called placodes. These placodes undergo remarkable morphological transformations to first invaginate as an epithelial bud and later morph into a more complex structure resembling the adult organ. Movement and rearrangement of epithelial cells are thought to drive tooth invagination. But there remain key knowledge gaps in our understanding of the underlying molecular mechanisms and signaling regulation that control cell movement and epithelial morphogenesis. Here, we use the developing mouse tooth as a model to show that Shh, a signaling molecule from the early tooth signaling center, plays an

**Funding:** This study was supported by the National Natural Science Foundation of China with grants NSFC 81900965 to WD and NSFC 82201003 to WD, by the Japan Society for the Promotion of Science KAKENHI with grant JP19K10047 to AY, and by the National Institute of Dental and Craniofacial Research (http://www.nidcr.nih.gov/) with grants R90DE031531 to QY, R01DE027620 and R35DE026602 to ODK, and R00DE025874 and R01DE030471 to JKH. The funders had no role in study design, data collection and analysis, decision to publish, or preparation of the manuscript.

**Competing interests:** The authors have declared that no competing interests exist.

important role in activating the motor protein non-muscle myosin II in dental epithelial cells. Myosin II is in turn required to promote strong adhesion between cells and power dental epithelial cells to converge towards the midline of the tooth bud to drive tooth invagination. Our results thus define the *in vivo* function of myosin II during epithelial invagination and explain how signaling from secreted molecules can facilitate morphogenetic cell movements via myosin II.

## Introduction

The formation of ectodermal organs, such as teeth, hair follicles, and glands, is a remarkable morphogenetic process that transforms simple epithelia into multilayered structures with distinct morphologies [1]. At the onset of this transformation, locally thickened epithelium, known as the placode, first stratifies and produces suprabasal cells that lie atop the basal layer. The epithelium then progressively bends and invaginates towards the underlying mesenchyme to form a bud shaped structure, which subsequently undergoes further growth, invagination, and organ-specific shape changes [2]. Epithelial invagination therefore represents a basic developmental module important for building epithelial organs, and disruption in this process underlies many genetic disorders that cause dysgenesis of ectodermal structures [3,4]. However, the underlying molecular and regulatory mechanisms involved in epithelial invagination are still poorly understood.

The mouse tooth has long served as a model for understanding epithelial morphogenesis [5]. Beginning at embryonic day 11 (E11), the dental placode emerges as a thickened stripe across the oral surface, marking the earliest morphological sign of tooth formation [6,7]. The dental placode then enlarges and invaginates into the underlying condensing mesenchyme, such that by E12.5 both incisor and molar buds are formed. A toothless space called the diastema separates these buds, as mouse dentition is reduced and lacks canines and pre-molars. Incisors and molars next undergo comparable morphological changes as they continue to invaginate into the mesenchyme, progressing through the cap and bell stages until the tooth erupts, although the incisor epithelium develops in an asymmetrical manner and turns posteriorly during the bud-to-cap transition at E13.5.

Through genetic studies in both mice and humans, we now know that many transcription factors and components of signaling pathways play critical roles during tooth development [8], but how they modulate cell behaviors to control morphogenesis is only beginning to be understood. Several recent studies have unveiled the well-coordinated patterns of cell proliferation and movement during the multi-step process of molar epithelial invagination. At the onset of epithelial bending, cellular protrusions at the apical surface are centripetally-oriented and push inner cells downwards to deform the epithelium [9]. In parallel, epithelial cells divide vertically to produce suprabasal cells, and Fgf signaling in the tooth germ is a driver of cell proliferation and dental placode stratification [10]. On the other hand, Shh, expressed in the early tooth signaling center, is not required for cell proliferation but for the subsequent narrowing of the dental cord and epithelial invagination [10]. Suprabasal cells of the invaginating epithelium are also motile, and they migrate towards the tooth midline while intercalating with one another [11]. Epithelial cells in the hair and mammary placodes undergo similar convergent movement, and cell motilities in these cases are influenced by Wnt and Shh, respectively—signaling pathways central to the development of these organs [12–14]. It was thus postulated that such a convergent cell movement generates the planar tissue tension observed in the suprabasal layer and is required during dental epithelial invagination to direct the downward

bending and growth of the placode towards the mesenchyme [15]. However, the underlying molecular mechanism that enables cell rearrangement and coordinates the collective directional movement to propel invagination is not clear.

In cells, tensile forces are primarily generated by the interaction between filamentous actin and the motor protein non-muscle myosin II (MyoII) [16]. Actomyosin tension transmitted through cell-cell adhesion can then organize cells into specific tissue shapes [17,18]. MyoII is composed of six subunits: two essential light chains, two regulatory light chains, and two heavy chains. Specifically, three isoforms of MyoII heavy chains are present in mice: IIA, IIB and IIC, which are encoded by *Myh9*, *Myh10*, and *Myh14*, respectively [19,20]. These three isoforms play both distinct and overlapping functions to regulate cell motility, arrangement, and adhesion during morphogenesis [21–24]. Interestingly, genetic mutations in regulators that function upstream to control actomyosin activity, including GTPase-activating proteins (GAPs) and small Rho GTPases, have been recently linked to dental anomalies in humans [25–29]. However, whether MyoII is required in the epithelium to control its invagination and shape is not understood, and how its activity may be integrated with upstream biochemical signals during tooth development is also to be addressed.

In this study, we investigate the functional requirement of MyoII in controlling the dynamic cellular organization in the developing mouse incisor bud. By combining explant culture and *in vivo* genetic studies, we found that MyoII is essential for tooth development and epithelial invagination. It does so by strengthening cell-cell adhesion at adherens junctions and by enabling efficient and persistent convergent cell movement in the suprabasal layer, where cells at the periphery of the tooth bud move towards the midline. Finally, we uncovered that Shh signaling acts through PI3K/AKT to activate MyoII. The Shh-PI3K/AKT-MyoII axis thus promotes proper cell-cell adhesion and cell movement to propel incisor invagination.

## Results

### Expression of myosin II in the developing incisor epithelium

We first assessed the expression pattern of all the MyoII heavy chain isoforms in the incisor epithelium by immunostaining at different developmental timepoints. From E12.5 to E13.5, when the incisor develops from an invaginating placode to a large bud, IIA and IIB are broadly expressed in both the epithelium and the mesenchyme, although IIA expression is more prominent in the epithelium (Fig 1A,1B,1E,1F,1I and 1J). In contrast, the expression of IIC is restricted to the periderm (Fig 1C,1G and 1K), the outermost layer of the epithelium that prevents inappropriate fusion between epithelial tissues. IIA and IIB are thus the main forms of MyoII heavy chains in the developing incisor. As MyoII activity depends on phosphorylation of the myosin regulatory light chain (pMLC), we next examined pMLC expression. Notably, upper suprabasal cells near the apical surface exhibited stronger pMLC expression than the lower and deeper suprabasal cells (Fig 1D,1H and 1L), thus indicative of higher MyoII activities and cellular forces in the upper suprabasal layer at these stages. pMLC is also strongly expressed in a few dividing cells (Fig 1L).

### Myosin II is required for proper incisor invagination

To begin investigating the functional role of MyoII in the mouse incisor, we utilized the mandible explant culture system and incubated dissected E12.5 mandibles in the presence or absence of the MyoII inhibitor blebbistatin for 2 days. Compared to control samples, explants treated with blebbistatin showed smaller and less invaginated incisor epithelium (S1A–S1C Fig). This was not caused by reduced cell proliferation, and MyoII inhibition even resulted in a slight increase in the percentage of BrdU-labelled cycling cells in the incisor germ when

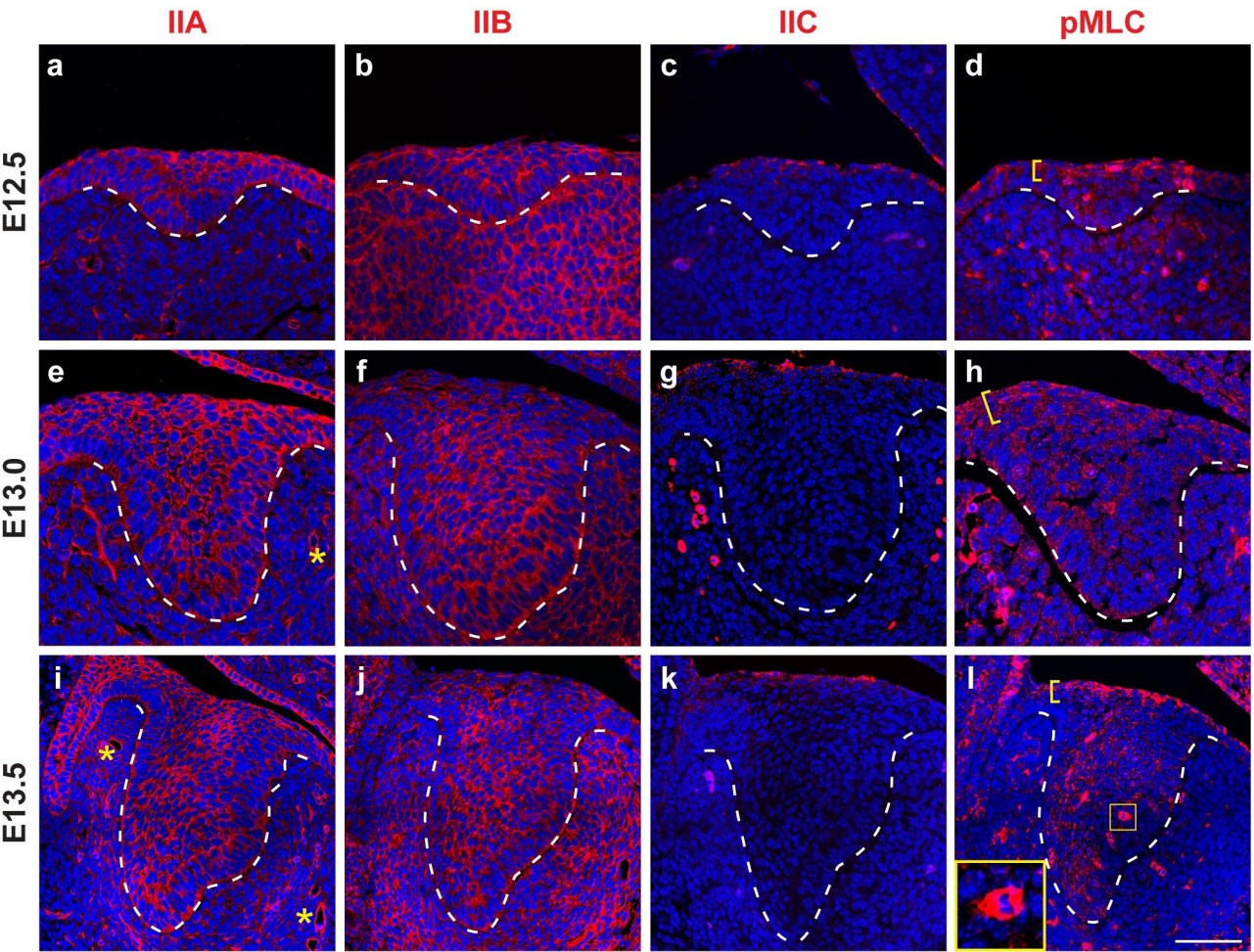

**Fig 1. Expression patten of myosin II in the early developing incisor epithelium.** (**a-l**) Immunostaining of non-muscle myosin heavy chains IIA, IIB, and IIC, as well as immunostaining of phospho-myosin light chain (pMLC) on sagittal sections of the developing incisor germ (anterior to the left) at E12.5 (a-d), E13.0 (e-h), and E13.5 (i-l). Brackets in (d, h, l) mark the stronger pMLC expression in the upper suprabasal cells. pMLC is also highly expressed in dividing cells (inset in l). Dashed lines outline the incisor epithelium. Asterisks mark IIA expression in the forming blood vessels. Representative images are shown. Scale bar in (l) represents 50 μm in (a-h) and 65 μm in (i-l).

compared to the control (S1D Fig). Mesenchymal proliferation remained unchanged (S1D Fig), indicating that under this experimental setting the epithelium and the mesenchyme responded differently to MyoII inhibition. The formation of the enamel knot in the incisor epithelium, as assessed by its marker *Shh*, was also not affected (S1E–S1G Fig). MyoII thus appears to promote tooth invagination without affecting proliferation and the establishment of the enamel knot signaling center.

To more concretely test if incisor epithelial invagination depends on the tissue-autonomous function of MyoII within the epithelium or its non-tissue-autonomous role in the mesen-chyme, we next turned to mouse genetic models. To ablate MyoII activity in the epithelium, we generated $K14^{CreER};R26^{mT/mG};Myh9/10^{f/f}$ embryos (henceforth called $Myh9/10^{epi-cko\ (epithelial\ conditional\ knockouts)}$), where the CreER recombinase under the control of a Keratin 14 promoter is expressed beginning at E11.0 and deletes both myosin IIA and IIB in the epithelium upon tamoxifen induction (S2A–S2G Fig). We did not target IIC, as it is only expressed in the peri-derm. $R26^{mT/mG}$ encodes a Cre reporter that permanently labels cells with membrane GFP

(mGFP) upon CreER recombination, allowing visualization of CreER-active cells and their progeny. Compared to CreER-negative control siblings, mutant embryos have considerably reduced myosin IIA and IIB expression in the dental epithelium at E12.5 and E13.5 (S2H–S2M Fig), confirming effective deletion in most cells. However, residual IIA and IIB can still be observed in some cells (S2H–S2M Fig), suggesting that a complete turnover of these proteins will sometimes take more than 48 hours.

Incisors with epithelial *Myh9/10* deletion initially formed normal placodes at E12.5 (S2H Fig), but akin to explants treated with blebbistatin, they later developed a less invaginated tooth germ with a wider dental cord at the neck region beginning at E13.5, when compared to the controls (Fig 2A–2C,2L and 2M). This phenotype became even more pronounced at E14.5, although the incisor's anterior-posterior morphological asymmetry was maintained (Fig 2D,2E,2L and 2M). Noticeably, the mutant tooth germs developed a visible cyst in the anterior upper suprabasal layer by E14.5 (n = 7/7; Fig 2E). We also examined the entire tooth germ by whole mount 2-photon imaging using $K14^{CreER};R26^{mT/mG}$ embryos as controls in order to visualize the epithelium using mGFP expression. Consistent with our observations in tissue sections, mutant incisors presented the same cyst phenotype and were quantifiably smaller in volume at both E13.5 and E14.5 (Fig 2H–2K and 2N). At E15.5, the mutant dental cord region remained wide and multiple holes were formed throughout the epithelium (Fig 2F and 2G). We also assessed the developing $Myh9/10^{epi-cko}$ molars from E13.5 to E15.5 and observed the same phenotypes (S3 Fig). MyoII proteins are thus required tissue-autonomously within the dental epithelium to maintain its integrity as well as to drive the morphological appearance of the dental cord and to enable effective invagination.

Because both myosin IIA and IIB are also expressed in the dental mesenchyme during epithelial invagination, we next generated $Msx1^{CreER};R26^{mT/mG};Myh9/10^{f/f}$ embryos ($Myh9/10^{mes-cko\ (mesenchymal\ conditional\ knockouts)}$) to test if mesenchymal deletion of IIA and IIB would impact the invagination of the dental epithelium (S2G Fig). Examining the incisor tooth germs at both E13.0 and E14.0 showed that the expression of IIA and IIB were decreased in the $Myh9/10^{mes-cko}$ mutant mesenchyme (S2N–S2S Fig). Yet both control and $Myh9/10^{mes-cko}$ incisor epithelia invaginated similarly (S2T Fig). While this result may be due to perdured IIB expression in some mesenchymal cells (S2P Fig), comparisons of the $Myh9/10^{epi-cko}$ and $Myh9/10^{mes-cko}$ mutants indicate that the tissue-autonomous function of epithelial MyoII is critical for proper tooth formation. We therefore focused on the $Myh9/10^{epi-cko}$ embryos and investigated further how epithelial MyoII controls incisor development.

## Loss of myosin IIA/B results in disruption of cell adhesion

The decrease in tooth size and invagination in $Myh9/10^{epi-cko}$ embryos could result from increased cell death and/or decreased cell proliferation. To probe these possibilities, we examined apoptosis and cell proliferation by means of TUNEL and EdU staining, respectively. When compared to control littermates, the mutant incisor epithelium did not show overt changes in apoptosis at E12.5 (12.5 ± 3.77 and 12.83 ± 7.69 TUNEL+ cells per section in controls and mutants respectively; n = 3). At E13.5, we did observe a slight, but significant, increase in the number of apoptotic cells in the region where the cyst began to form (S4A–S4C Fig). Cell death could therefore contribute to the cyst formation. However, cell death alone is unlikely to be the main cause of the invagination phenotype observed, as tooth germs with a higher amount of apoptosis still invaginated normally in other genetic mutant mouse embryos that we have worked with, for example with epithelial deletion of *Piezo 1* using $K14^{CreER}$ (S5 Fig). We next assessed cell proliferation and found that there was a small, but not statistically significant, increase in the percentage of EdU+ proliferating cells in the basal and the

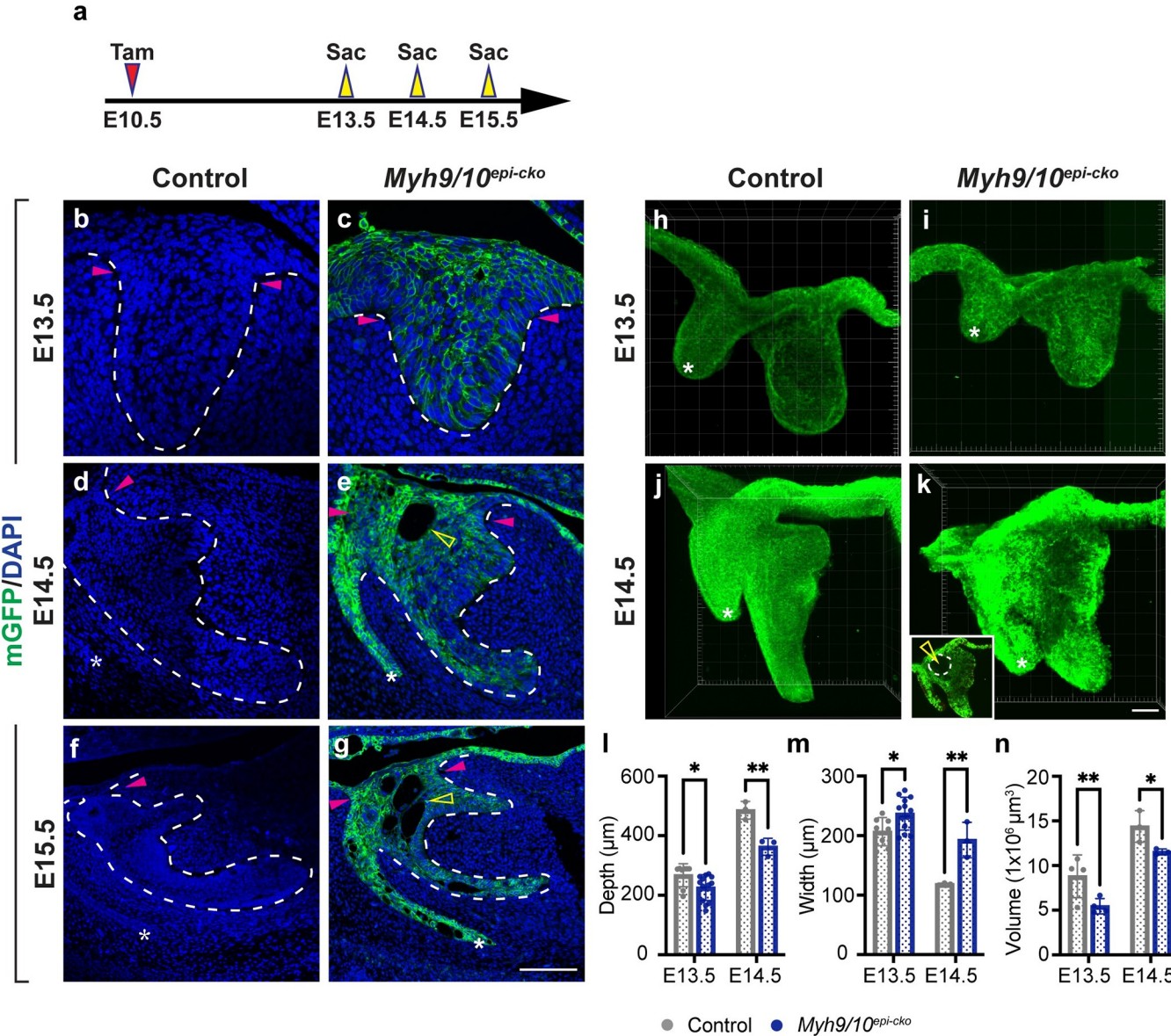

**Fig 2. Myosin II is required for proper incisor invagination.** (**a**) Timeline depicting the onset of CreER induction by tamoxifen (Tam) through oral gavage at E10.5 (red arrowhead) and sample collections at E13.5-E15.5 (yellow arrowheads). (**b-g**) Sagittal sections through control (CreER-negative) and mutant (*Myh9/10*$^{epi-cko}$) incisors at E13.5, E14.5, and E15.5 show reduced invagination and a wider dental cord in the mutant incisor epithelium. Pink arrowheads denote the dental cord region. Open yellow arrowheads in (e and g) mark the cysts that are formed in mutant incisors. mGFP is a Cre-reporter. (**h-k**) Whole mount 2-photon images of control and *Myh9/10*$^{epi-cko}$ incisor epithelia at E13.5 and E14.5. Inset in (k) is a single z-section and the open yellow arrowhead marks the cyst. (**l** and **m**) Quantifications of the epithelial depth and the neck width in control and *Myh9/10*$^{epi-cko}$ incisors at E13.5 (n = 8 controls and 13 mutants) and E14.5 (n = 3 controls and 3 mutants). (**n**) Quantification of the incisor epithelial volume at E13.5 (n = 4 per genotype) and E14.5 (n = 3 per genotype). Dashed lines outline the incisor epithelium. Asterisks (*) mark the tip of the vestibular lamina. Representative images are shown. All quantitative data are presented as mean ± SD. The p values were determined using an unpaired Student's t-test (* p < 0.05 and ** p < 0.01). Scale bar in (g) represents 50 µm in (b and c), 90 µm in (d and e) and 125 µm in (f and g), scale bar in (k) represents 50 µm in (h-k).

suprabasal layer of the mutant incisors at E13.5 (S4D–S4F Fig). This trends similarly to results from the blebbistatin culture experiment (S1D Fig), and variations in the proliferation amounts and patterns are likely due to differences between experimental systems. The initiation knot and the enamel knot in *Myh9/10*$^{epi-cko}$ incisors were also formed correctly based on the expression of their marker gene, *Shh*, and the Hedgehog signaling-responding gene, *Ptch1*

(S4G–S4P Fig). These signaling centers' role in supporting cell proliferation and tooth development would thus remain intact. Together, these results indicate that MyoII is dispensable for maintaining proper signaling center formation and dental epithelial proliferation, and the invagination phenotype is not due to perturbations or a delayed onset in these developmental events.

Because earlier reports have shown that defective cell-cell adhesion could cause cystic changes during odontogenesis [30,31], we next tested if cell-cell adhesion was affected in the absence of MyoII activity. We first examined the expression of the adherence junction protein E-cadherin (E-cad) at E13.0, a stage prior to the formation of cysts, by immunostaining. While the total expression level of E-cad was not affected in the mutant tooth germs at this stage (Fig 3A–3C), detection of active E-cad using the ECCD2 antibody that only recognizes the extracellular domain of E-cad under homophilic interactions [32] showed decreased E-cad clustering in mutants at E13.0 (Fig 3D–3E' and 3H). Consistent with this result, β-catenin, the intracellular binding partner of cadherins and an important promoter of cadherin clustering and cell-cell adhesion strength [33], was also significantly diminished at the cell membrane at E13.0 (Fig 3F–3G' and 3I). These results are consistent with previous findings that MyoII activity is required for maintaining strong cell-cell adhesion at adherence junctions [34,35], and weakened adhesion in the $Myh9/10^{epi-cko}$ epithelium likely contributes to the subsequent cyst formation.

## Deletion of MyoII disrupts cellular organization and movement during incisor invagination

We next set out to understand how MyoII controls incisor morphogenesis, as the $Myh9/10^{epi-cko}$ tooth germ is wider at the epithelial cord region and has reduced invagination beginning at E13.5 (Fig 2). Because cell and nuclear morphology can be influenced by cell-cell adhesion and tissue forces and has predictive power for collective cell motility [36,37], we first compared nuclear shapes in the upper suprabasal layer between control and $Myh9/10^{epi-cko}$ mutant embryos at E13.0. This showed that $Myh9/10^{epi-cko}$ cells had rounder, less elongated nuclei with smaller aspect ratios (major axis divided by minor axis of a fitted ellipse) (Fig 4A–4D), reflective of reduced cellular tension. Correspondingly, actin intensity by phalloidin staining was also weakened in the mutants at E13.0 (Fig 4E–4G). Closer examination of the phalloidin labelling revealed that filamentous actin was typically concentrated as bright foci at the tricellular regions, where intercalation would be taking place (Fig 4H). This is reminiscent of the actin localization in the *Xenopus* gastrula mesoderm during its convergent extension [38], where actomyosin-mediated cell-cell adhesion at the tricellular interface facilitates cell intercalation. However, in the mutant incisor germs many of these foci were lost (Fig 4I and 4J), suggesting a defect in cellular interactions and convergence.

To test how MyoII is required for proper cellular movement in the invaginating incisor, we then performed live imaging of mandibular sagittal slices. We needed to first establish how cells normally move and behave in the control incisor epithelium and used E13.0 $K14^{CreER}$; $R26^{mT/mG}$ embryos to track cell divisions and cell movements. We focused on this timepoint because it precedes the $Myh9/10^{epi-cko}$ invagination defect at E13.5 and the cyst formation at later stages. Over 6–8 hours of imaging, mitosis was readily observed in both the suprabasal and the basal layers of cultured control $K14^{CreER};R26^{mT/mG}$ incisor epithelia, indicating that cells remained active in the culture environment. The orientations of these divisions largely followed Hertwig's rule, which states that an elongated cell divides along the long axis of the cell [39]. For example, the upper suprabasal cells were elongated antero-posteriorly and they divided in the planar orientation (S1 Movie). This directional division could also result from

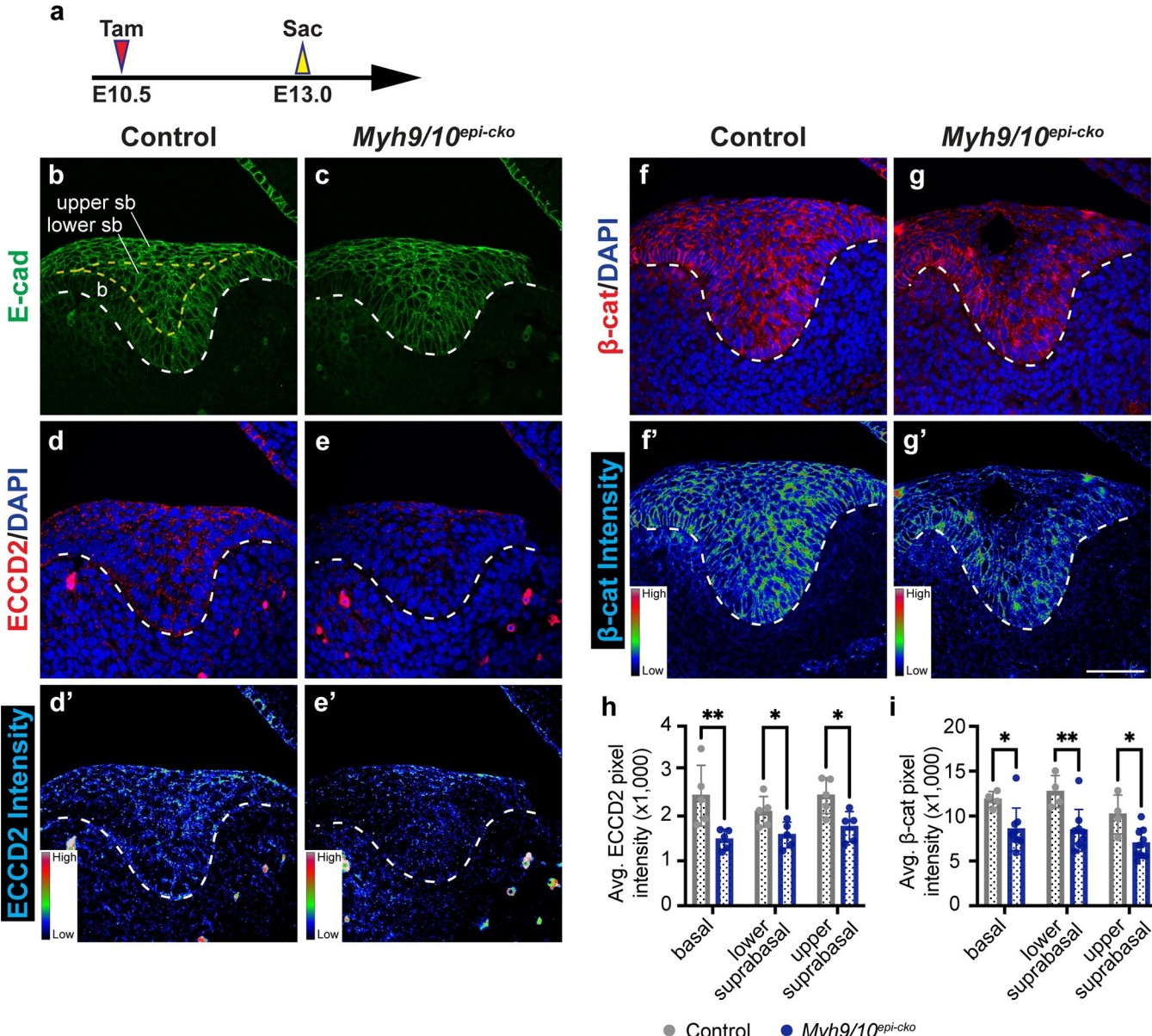

**Fig 3. Myosin II is essential for cell adhesion in the incisor epithelium.** (**a**) Timeline depicting the onset of CreER induction by tamoxifen (Tam) through oral gavage at E10.5 (red arrowhead) and sample collections at E13.0 (yellow arrowhead). (**b-g**) Immunostaining of total E-cadherin (E-cad) (b and c), homophilically bound E-cad (ECCD2 in d and e), and β-catenin (β-cat) (f and g) in control and *Myh9/10*$^{epi-cko}$ samples. (**d'-g'**) The corresponding signal intensity heatmap for ECCD2 (d' and e') and β-catenin (f' and g'). (**h** and **i**) Quantifications of the average signal intensity per pixel for ECCD2 (h) (n = 5 per genotype) and β-cat (i) (n = 4 controls and 9 mutants) immunostaining in different sub-regions of the incisor epithelium. The basal (b), lower suprabasal (sb) and upper suprabasal sub-regions are determined as shown in (b). Dashed lines outline the incisor epithelium. Representative images are shown. All quantitative data are presented as mean ± SD. The p values were determined using an unpaired Student's t-test (* $p < 0.05$ and ** $p < 0.01$). Scale bar in (g') represents 50 μm in (b-g').

the planar contractile forces present in the dental suprabasal layer [11], as tissue tension can be a key determinant of the division orientation [40,41]. On the other hand, the columnar basal cells often divided vertically relative to the basement membrane (S1 Movie), and the more apically positioned daughter cells could then either enter the suprabasal layer or reinsert back to the basal layer (S2 Movie, S3 Movie, and S4 Movie). At the ventral apex of the tooth bud, we

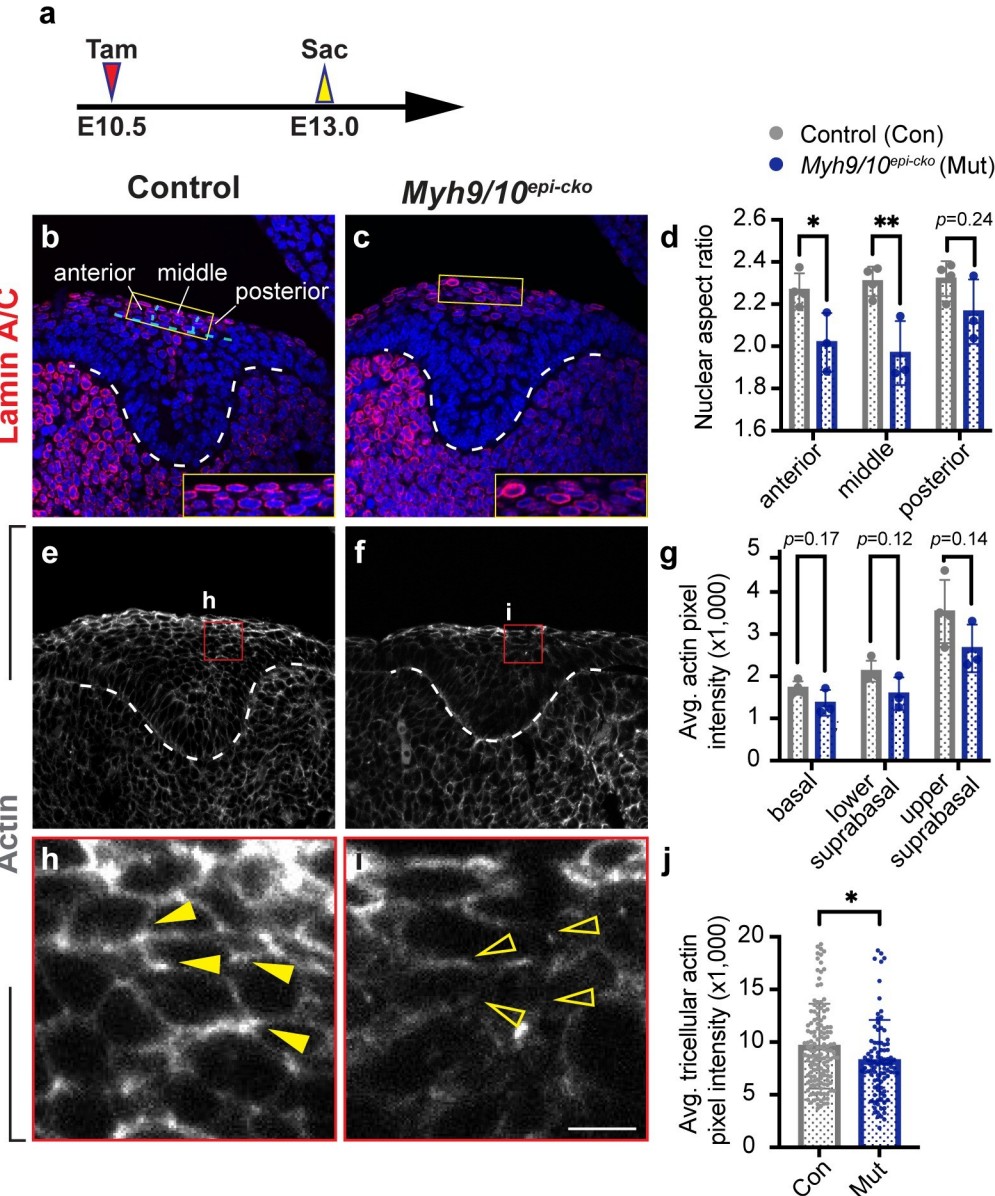

**Fig 4. Deletion of myosin II alters nuclear shapes and actin distribution in the upper suprabasal layer.** (**a**) Timeline depicting the onset of CreER induction by tamoxifen (Tam) through oral gavage at E10.5 (red arrowhead) and sample collections at E13.0 (yellow arrowhead). (**b-d**) Lamin A/C immunostaining (b and c) traces nuclear shapes in the upper suprabasal layer in control and *Myh9/10epi-cko* incisors; anterior to the left. Nuclear aspect ratios are quantified (d) in the anterior, middle, and posterior parts of the upper suprabasal region, as indicated in (b). (n = 4 controls and 3 mutants). (**e-j**) Phalloidin staining of actin (e and f) in control and *Myh9/10epi-cko* incisors. Areas marked by the red squares are enlarged in (h and i), showing actin enrichment at the tricellular vertices (yellow arrowheads) of control cells and the loss of tricellular actin foci (open yellow arrowheads) in *Myh9/10epi-cko* mutant cells. The average actin pixel intensity (n = 4 controls and 3 mutants) and the average tricellular actin pixel intensity (n = 140 foci from 4 controls and 96 foci from 3 mutants) are quantified (g and j). Dashed lines outline the incisor epithelium. Representative images are shown. All quantitative data are presented as mean ± SD. The *p* values were determined using an unpaired Student's *t*-test (* *p* < 0.05 and ** *p* < 0.01). Scale bar in (i) represents 50 μm in (b-c, e-f).

also observed occurrences of horizontal divisions, after which both daughter cells would remain in the basal layer (S4 Movie and S5 Movie). Finally, basal cells could directly delaminate into the suprabasal layer (S4 Movie). We next focused on cell movements that could drive

epithelial invagination in these control tissues. At the shoulder junctions of the dental epithelium that bridge the incisor bud to the adjacent oral epithelium, the adjoining suprabasal and basal cells often remained closely associated with each other, even after the formation of suprabasal cells from the basal layer (Fig 5A,5B and 5E). The cell bodies of these lateral suprabasal cells extended centripetally and interdigitated with more medially located suprabasal cells. The cohort of basal and lateral suprabasal cells then collectively moved as a unit towards the midline of the tooth germ, with the posterior cells exhibiting more pronounced movement than the anterior cells (Figs 5C and 5E, S6A–S6H and S6 Movie). The more centrally located cells also underwent intercalation, and as they converged towards the midline, they frequently displaced the middle cells sandwiched in between away in the z axis (Figs 5D, S6C, S6G and S6 Movie). Together, these collective convergent cell movements would accompany, on average, 46% narrowing of the incisor neck region and 26% lengthening in the epithelial depth over the course of the explant culture (Fig 5K and 5L), thus facilitating epithelial invagination in a manner that is similar to the process previously described in molars [11].

In contrast, the mutant incisor epithelium did not narrow to the same extent at the neck region as their control counterparts during explant culturing, and epithelial invagination in mutants was also reduced (Fig 5K and 5L). But the cell proliferation pattern appeared normal during live imaging. To investigate further the underlying cause of decreased invagination, we tracked cell movement towards the midline of the tooth germ and quantified both cell velocity (total distance travelled divided by time) and efficiency (displacement divided by total distance travelled) in both control and mutant incisors at E13.0 (Figs 5A–5J, 5M–5O and S6, S6 and S7 Movies). We focused on the posterior incisor epithelium, as cells in that region had the most pronounced movement and we wanted to avoid potential complications caused by the forming cysts anteriorly that could alter the path of moving cells. Interestingly, we found that mutant cells in the *Myh9/10*$^{epi-cko}$ posterior incisor buds were not statistically slower than cells in the control epithelia, but they moved in a less efficient manner (Fig 5M and 5N). Quantifying the turning angles between each imaged movement as well as the spread of individual cell's total turning angles further indicates that mutant cells tended to switch directions and go backwards, while control cells moved both more linearly and persistently towards the midline (Fig 5O–5Q). These results thus revealed that MyoII is required for cells to converge towards the tooth midline in a directionally persistent manner that is coupled with suprabasal narrowing. In the absence of MyoII, cells moved more randomly, and the epithelium could not efficiently narrow to drive invagination. Finally, we also noticed that as control tooth germs invaginated, suprabasal cells became aligned around the forming enamel knot at E13.5. However, *Myh9/10*$^{epi-cko}$ cells are more disorganized and showed reduced coherency in cell orientations (S7 Fig). MyoII is therefore critical for organized cellular movement and arrangement during epithelial invagination.

## Shh signaling acts through AKT and MyoII to promote cell adhesion and epithelial invagination

We next investigated the upstream signal that activates MyoII activity in the invaginating incisor. Between E12.5 and E13.5, Shh is a key signaling molecule expressed in the initiation knot signaling center of the developing tooth, and Hedgehog signaling activity can be readily detected in the entire dental epithelium, as assessed using the expression of its target gene *Ptch1* (Fig 6A) [42–44]. Importantly, previous studies have shown that deletion of *Shh* resulted in a shallower tooth germ with a wide neck region [45], and blockade of Shh activity by cyclopamine in cultured explants similarly inhibited molar invagination [10]. The phenotypic resemblance of *Shh* and *Myh9/10*$^{epi-cko}$ mutants thus prompted us to hypothesize that Shh

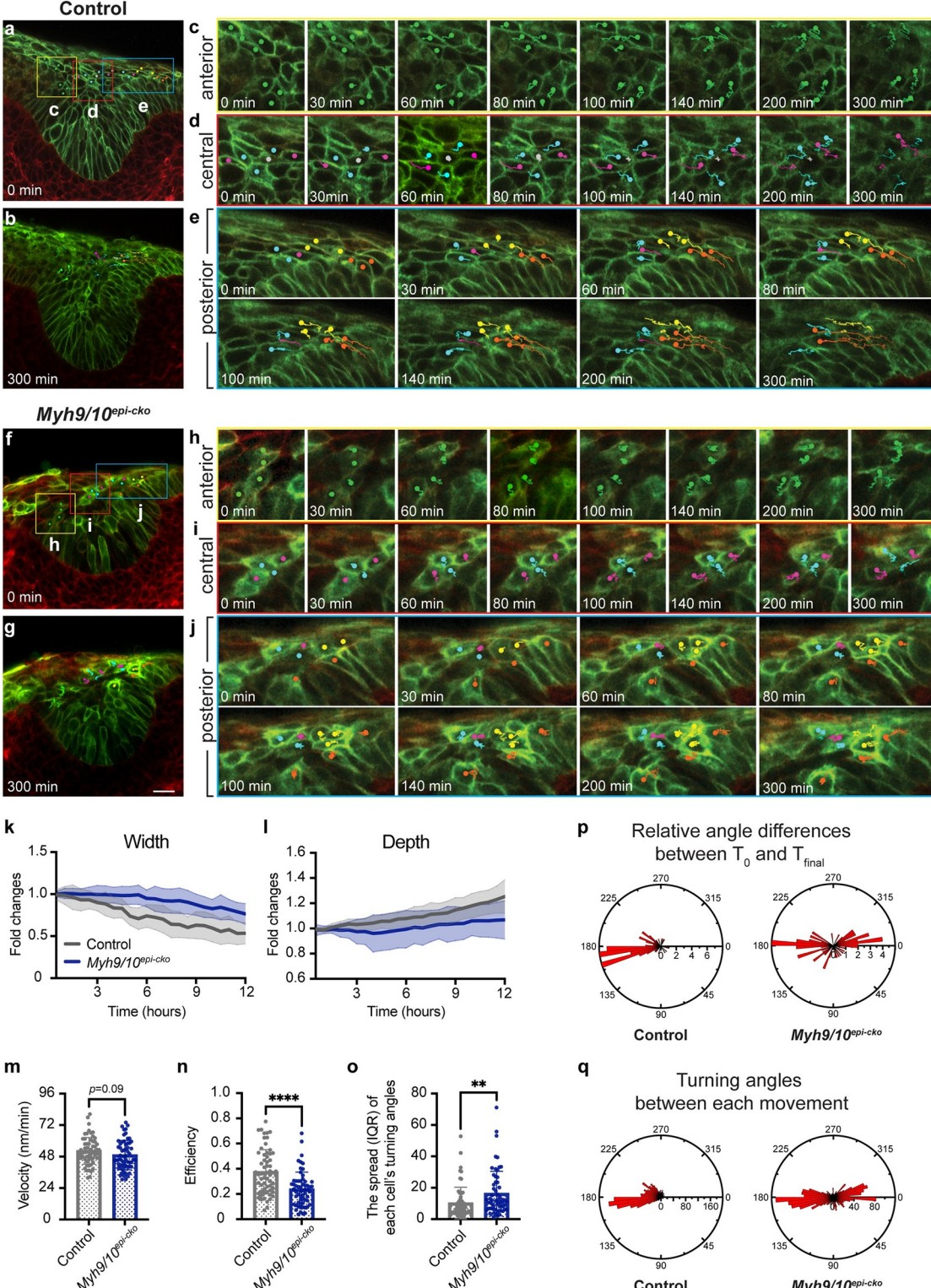

**Fig 5. Myosin II is required for efficient convergent cell movement during incisor invagination.** (**a–j**) Two-photon time-lapse live imaging shows the tracked movement of E13.0 control (a-e) and mutant (f-j) epithelial cells from the anterior (yellow squares), central (red squares), and posterior (cyan squares) regions of the upper incisor germ. Green dots mark anterior cells. In the control tissue, pink dots mark cells converging towards the midline, displacing cells marked by cyan dots. Gray dots mark cells displaced in the z-planes. Orange dots mark posterior basal cells and yellow dots mark adjacent co-migrating suprabasal cells. These movements

are disrupted in myosin II-deleted mutants. Cell membranes are labelled by the dual color CreER-reporter mTmG; green indicating CreER-mediated recombination and red indicating no prior CreER activity. (**k** and **l**) Fold changes of the epithelial depth and neck width of control or $Myh9/10^{epi-cko}$ incisors over time. (n = 8 controls and 7 mutants). (**m-o**) Quantifications of control and mutant cell velocity (m), efficiency (n), and the spread (measured in interquartile range or IQR) of each cell's turning angles between movements from the start to the end of the time-lapse (o). Each dot represents a single tracked cell. (n = 62 control cells from 8 control embryos and 60 mutant cells from 7 mutant embryos). (**p**) Rose diagram showing cells' relative positional changes between the beginning ($T_0$) and the end ($T_{final}$) of the time-lapse. (**q**) Rose diagram showing all cells' turning angles between each tracked cellular movement. Representative images are shown. All quantitative data are presented as mean ± SD. The *p* values were determined using an unpaired Student's *t*-test (** $p < 0.01$ and **** $p < 0.0001$). Scale bar in (g) represents 20 μm in (a, b, f, g).

functions upstream of MyoII to control tooth invagination. To test this, we treated cultured E12.5 mandibles with cyclopamine or control vehicle ethanol for 24 hours and then examined phenotypic changes in the incisor epithelium (Fig 6B). Inhibition of Hh signaling, as expected, downregulated *Ptch1* expression (S8A,S8B,S8D Fig), but it also attenuated the levels of pMLC (Fig 6C–6D' and 6G) and active RHOA (S9A–S9B',S9D Fig), a small GTPase that activates MLC through Rho-associated protein kinase (ROCK) [16]. Because Shh has been shown to activate RHOA by signaling through PI3K/AKT in a G protein-dependent non-canonical pathway [46–48], we then performed phospho-AKT (pAKT) immunostaining and found a corresponding decrease when Hh signaling was blocked (Fig 6H–6I' and 6L). These results thus position Shh as an upstream regulator of AKT and MyoII.

Morphologically, the cyclopamine-treated incisors invaginated less than the controls (Fig 6M and 6N), but led by our findings above, we predicted that forced activation of the PI3K/AKT-MyoII signaling axis could rescue the invagination phenotype. To that end, we performed MyoII rescue by incubating E12.5 explants with both cyclopamine and calyculin A, an MLC phosphatase inhibitor that can activate MyoII activity [49]. This restored pMLC expression without significantly affecting pAKT in the epithelium after 24 hours of culture, especially in the more accessible suprabasal layer (Fig 6E,6E',6G,6J,6J' and 6L). To rescue AKT and PI3K, we used SC79 and UCL-TRO-1938 respectively [50,51], and they each re-established the levels of pAKT, as well as pMLC and active RHOA, in cyclopamine-treated samples (Figs 6F,6F',6G,6K,6K',6L and S9C–S9F',S9I,S9J). In all instances, the activation of MyoII or AKT was able to rescue the invagination defect caused by Shh inhibition, as the rescued epithelial tissues exhibited a similar depth and width-to-depth ratio as the control samples (Figs 6M and 6N and S9M,S9N). We also performed MyoII rescue in older tooth germs using E13.5 mandibular tissue slices and obtained the same result (S10 Fig). It should be noted that calyculin A could potentially rescue the invagination phenotype by activating Shh signaling itself, as phosphatase inhibition by calyculin A has been shown to maintain phosphorylation and activation of Smoothened (Smo) [52,53]. We therefore measured *Ptch1* expression in control and drug-treated samples to test this alternative explanation. However, no rescue of *Ptch1* expression was observed (S8C,S8D Fig), indicating that calyculin A was not capable of re-activating Smo in the presence of cyclopamine. The result is consistent with the known function of cyclopamine to lock Smo in a closed conformation by blocking its phosphorylation sites [54], thus leaving it impervious to calyculin A-induced activation. Together, our findings here show that Shh can signal through PI3K/AKT to activate MyoII and drive dental epithelial invagination.

We next asked if Shh functions through MyoII to promote cell adhesion and convergent movement. Akin to the $Myh9/10^{epi-cko}$ mutants, cyclopamine-treated incisors displayed reduced ECCD2 and β-catenin expression (Fig 7A–7B',7E,7F–7G',7J). There was no apparent cyst formation, which may require a prolonged culture time to develop and/or a more complete loss of adhesion. We also tracked cell movement by live-imaging E13.0 mandible slices after a 3-hour incubation with either control vehicle (ethanol) or cyclopamine. This revealed that posterior cells in the cyclopamine-treated incisors failed to move efficiently towards the

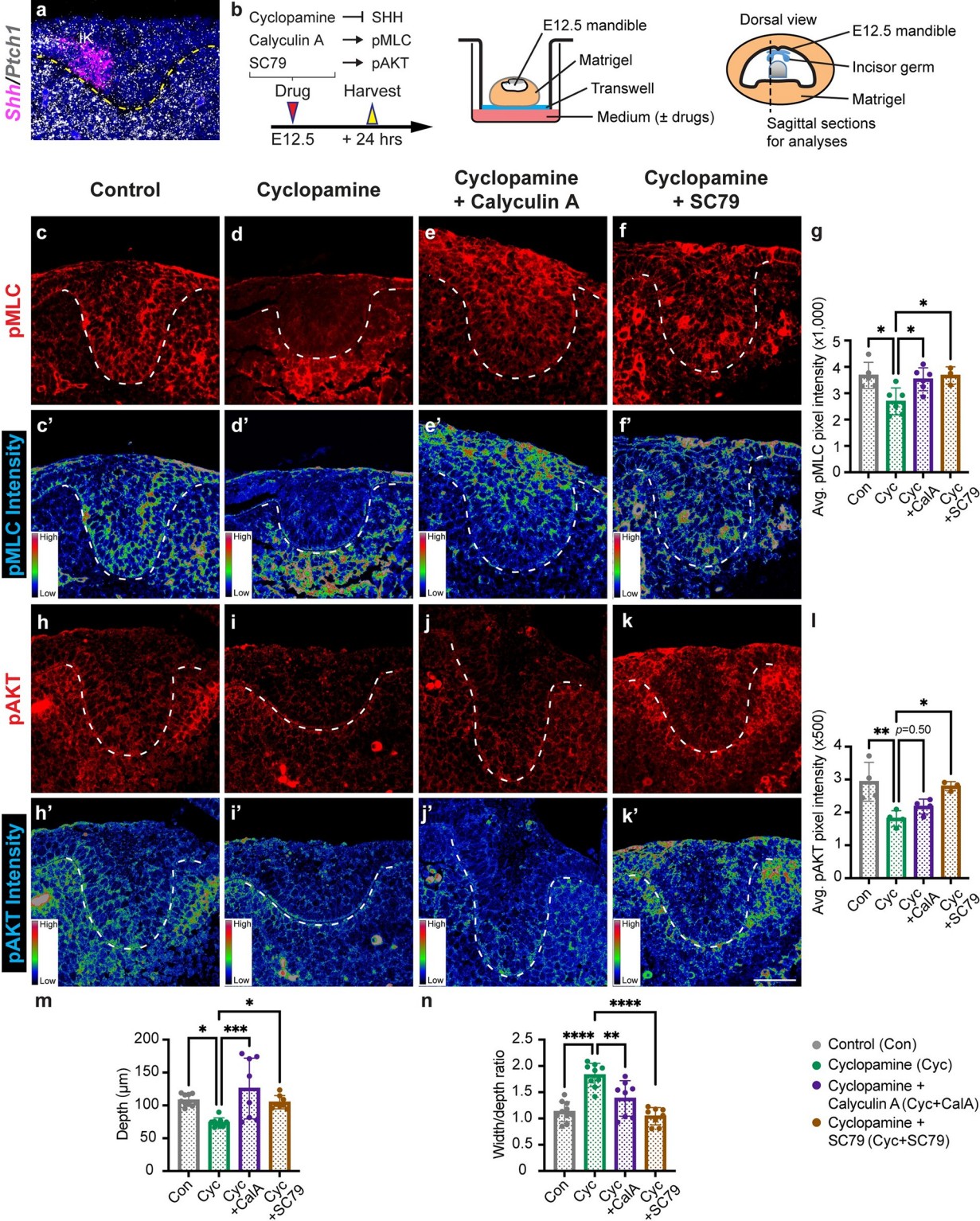

**Fig 6. Shh regulates dental epithelium invagination via AKT and myosin II.** (**a**) At E12.5, *Shh* is expressed in the incisor signaling center, the initiation knot (IK), and *Ptch1* is expressed in the surrounding epithelium and mesenchyme; anterior to the left. (**b**) Schematic of the explant culture system and the pharmacological agents used. (**c-g**) Immunostaining of pMLC (c-f) in control, Shh-inhibited (cyclopamine), MyoII-rescued (cyclopamine plus calyculin A) and AKT-rescued (cyclopamine plus SC79) tooth germs. The average pMLC signal intensity per pixel is quantified for all four conditions (g) (n = 5, 6, 6, and 3 respectively). (**c'-f'**) The corresponding pMLC signal intensity heatmap for (c-f). (**h-l**)

Immunostaining of pAKT (h-k) in control, Shh-inhibited (cyclopamine), MyoII-rescued (cyclopamine plus calyculin A) and AKT-rescued (cyclopamine plus SC79) tooth germs. The average pAKT signal intensity per pixel is quantified for all four conditions (l) (n = 4, 4, 4, and 3 respectively). (**h'-k'**) The corresponding pAKT signal intensity heatmap for (h-k). (**m** and **n**) Quantifications of the incisor invagination depth (m), as well as the width to depth ratio (n) in control (n = 10), Shh-inhibited (cyclopamine, n = 9), MyoII-rescued (cyclopamine plus calyculin A, n = 8) and AKT-rescued (cyclopamine plus SC79, n = 8) samples. Dashed lines outline the incisor epithelium. Representative images are shown. All quantitative data are presented as mean ± SD. The $p$ values were determined using one-way ANOVA and Tukey's HSD test for g, l, m, and n. (* $p < 0.05$, ** $p < 0.01$, *** $p < 0.001$, **** $p < 0.0001$). Scale bar in (k') represents 37 μm in (a) and 50 μm in (c-f', h-k').

midline of the tooth bud (Fig 7K–7L",7O, S8 and S9 Movies), recapitulating the cell movement defect seen in *Myh9/10$^{epi-cko}$* mutants. Remarkably, both the adhesion and cell movement phenotypes were significantly rescued when MyoII activity was restored using calyculin A or 4-hydroxyacetophenone (4-HAP) [55], and when AKT was re-activated using SC79 or UCL-TRO-1938 (Figs 7C–7E,7H–7J,7M–7O, S9G–S9H',S9K,S9L, S11, S10 and S11 Movies). Results from these experiments thus support a model, in which Shh signals through PI3K/AKT and MyoII in the tooth epithelium to maintain proper cell-cell adhesion and to promote convergent cell movement that drives epithelial invagination.

## Discussion

Epithelial invagination is a fundamental morphogenetic process that transforms an initially flat sheet into a bent and more complex three-dimensional structure during the development of many epithelial organs. While biochemical signaling events are important regulators of this process [56,57], it is less clear how signaling pathways and cells' mechanical machineries are coupled to direct specific cell arrangements and movements during invagination. Here, we use the developing mouse incisor as a model and identify Shh as an upstream promoter of MyoII activation. MyoII in turn maintains strong cell-cell adhesion and enables directionally persistent cell movement from the edge of the tooth germ towards the placode center. The ensuing convergent cell movement narrows the apical layers of the epithelium, effectively directing further epithelial growth and invagination towards the underlying mesenchyme.

### The role of myosin II in cell proliferation and cell death in the incisor epithelium

Although MyoII has been shown to exert tension at the contractile ring of dividing cells and is required for cytokinesis in both cultured cells and in some tissues, including the heart [58–60], whether it controls cell divisions in the developing tooth was not clear. We found that in the absence of myosin IIA and IIB, cell proliferation can still take place, and divisions were observed throughout the epithelium in live-imaged incisor buds. The requirement of MyoII in driving proper cell divisions is therefore context dependent. In fact, the percentage of proliferating cells were slightly elevated in the upper suprabasal region of the E13.5 *Myh9/10$^{epi-cko}$* mutant incisors, as well as blebbistatin-treated explants. This is similar to findings in other epithelial systems, where loss of myosin IIA and/or IIB also resulted in increased proliferation [61,62]. Epithelial cells are inherently adherent and adherence junction-mediated contact inhibition plays an important role in modulating epithelial cell proliferation [63]. For example, deletion of α-catenin or p120-catenin, both adherence junction proteins, led to increased epithelial proliferation [64,65]. Consequently, the subtle proliferation increase in the MyoII mutants could result from the loss of cell-cell adhesion. Suprabasal cells surrounding the forming cyst at E13.5 may also elevate proliferation in order to compensate for the tissue loss [66]. Lastly, we considered whether cell death is responsible for the cyst formation and the reduction in incisor invagination. However, apoptosis was not increased prior to cyst formation and likely secondary to lost adhesion. Apoptosis alone is also unlikely to be the main cause of the

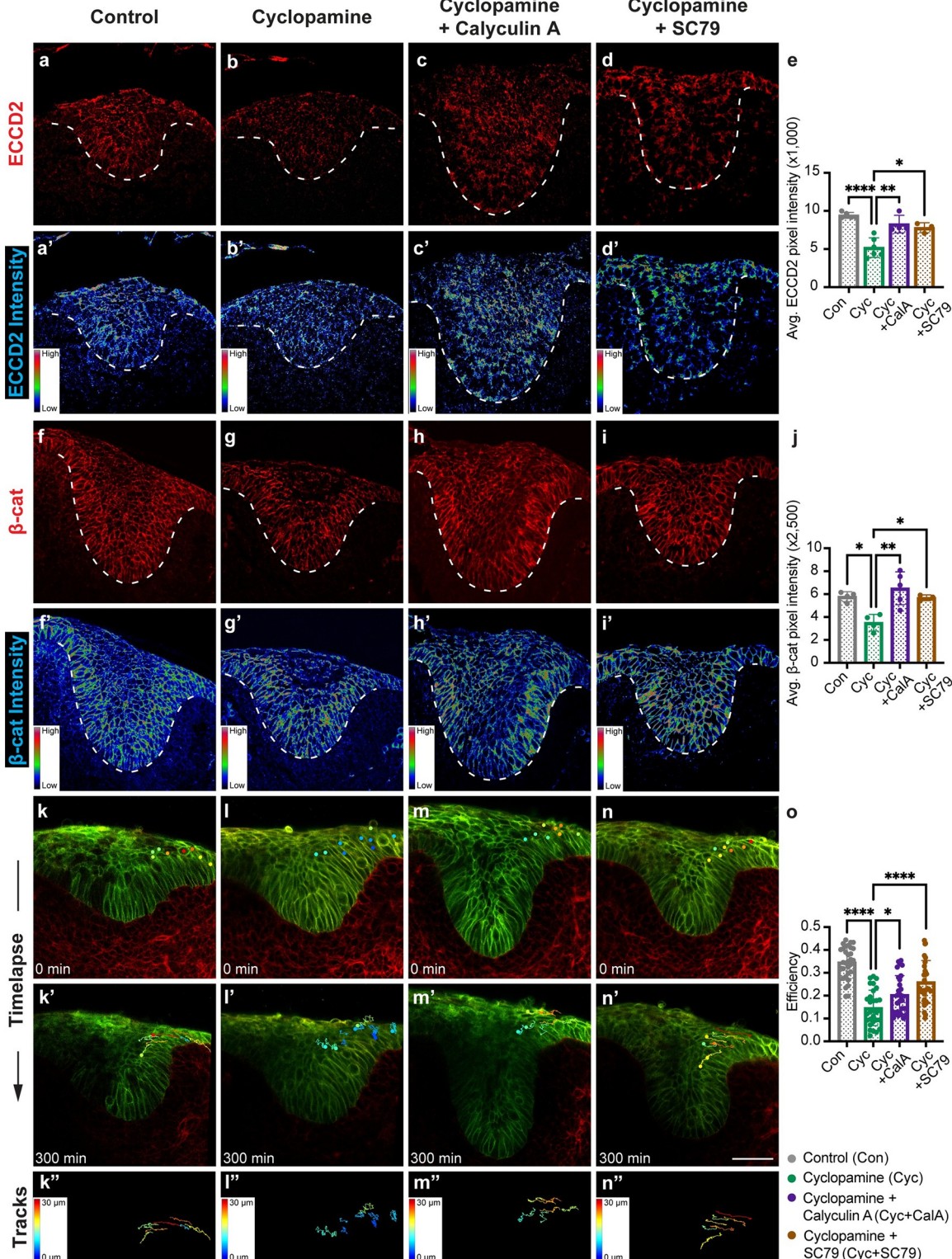

**Fig 7. Shh regulates dental epithelial cell adhesion and movement via AKT and myosin II.** (**a-e**) Immunostaining of homophilically bound E-cad (ECCD2) (a-d) in control, Shh-inhibited (cyclopamine), MyoII-rescued (cyclopamine plus calyculin A) and AKT-rescued (cyclopamine plus SC79) samples. The average ECCD2 signal intensity per pixel is quantified for all four conditions (e). (n = 5, 6, 4, and 3 respectively). (**a'-d'**) The corresponding ECCD2 signal intensity heatmap for (a-d). (**f-j**) Immunostaining of β-catenin (β-cat) (f-i) in control, Shh-inhibited (cyclopamine), MyoII-rescued (cyclopamine plus calyculin A) and AKT-rescued (cyclopamine plus SC79) tooth

germs. The average β-catenin signal intensity per pixel is quantified for all four conditions (j). (n = 4, 4, 5, and 3 respectively). (**f'-i'**) The corresponding β-cat signal intensity heatmap for (f-i). (**k-o**) Time-lapse live imaging shows the tracked movement of dental epithelial cells from the posterior regions of the incisor germ in control (k-k"), Shh-inhibited (cyclopamine) (l-l"), MyoII-rescued (cyclopamine plus calyculin A) (m-m"), and AKT-rescued (cyclopamine plus SC79) (n-n") samples. Track colors correspond to the distance of cell displacements. The efficiency of cell movement is quantified in (o). Each dot represents a single tracked cell. (n = 30, 30, 28 and 28 cells respectively from 3 embryos in each culture condition). Cell membranes are labelled by the dual color Cre-reporter mTmG; green indicating Cre-mediated recombination and red indicating no prior Cre activity. Dashed lines outline the incisor epithelium. Representative images are shown. All quantitative data are presented as mean ± SD. The $p$ values were determined using one-way ANOVA and Tukey's HSD test for e, j, and o. (* $p < 0.05$, ** $p < 0.01$, **** $p < 0.0001$). Scale bar in (n') represents 50 μm in (a-d', f-i') and 45 μm in (k-n').

invagination defect, as incisor buds with extensive cell death could still invaginate properly (S5 Fig). It remains to be determined whether a combined effect from both cell death and cyst formation can impede tooth invagination. As cell-cell connections normally persist between adjacent basal and suprabasal cells during invagination [11], losing such connections at a large scale due to the presence of a cyst could disrupt tissue force transmission in MyoII mutants and further add to the invagination phenotype after E13.5.

## Control of epithelial adhesion and invagination by myosin II

In the absence of myosin IIA and IIB, our analysis found that cell-cell adhesion through adherence junctions is disrupted in the invaginating incisor epithelium. While loss of MyoII did not affect the expression of total E-cadherin, it significantly reduced the amount of homophilically bound E-cadherin and membrane-localized β-catenin in the suprabasal layer. The weakened adhesion would then eventually contribute to impaired tissue integrity and the formation of cysts. Our results thus demonstrate the importance of MyoII in maintaining robust cell-cell adhesion in the stratified dental epithelium. This is also consistent with prior findings showing that MyoII-generated cellular forces are essential for the maintenance and the growth of adherence junctions in cultured cells [67,68], and deletion of myosin IIA and/or IIB results in mislocalized E-cadherin and defective adhesion in mouse embryos [34,69].

Maintaining adequate intercellular adhesions is also important for collective cell movement, such as during convergent extension [70]. In this context, adhesion provides the anchoring needed to organize actin assembly and actomyosin tension at the tricellular interface, where cell crawling and cell-cell junction shortening are integrated to achieve effective cell intercalation [38]. Similarly in the developing incisor, where we have observed convergent extension cell movement in the suprabasal layer, strong actin accumulation was detected at the tricellular region. However, with reduced adhesion in the E13.0 $Myh9/10^{epi-cko}$ mutant incisors, tricellular actin also became delocalized and effective cell interaction was compromised.

Using live imaging, we showed that even before any cyst formation $Myh9/10^{epi-cko}$ mutant incisors already developed a clear cell movement phenotype at E13.0, in which posterior suprabasal cells often reciprocated and displayed reduced efficiency of convergent cell movement. The result is a wider neck at the dental cord region and a shallower tooth bud in the MyoII mutants. In contrast, suprabasal cells in the posterior control incisors persistently moved towards the tooth midline while intercalating and displacing medially located cells. Surprisingly, given the role of MyoII in controlling migration [16], deletion of IIA/B did not abolish cell motility in the incisor, and cell velocities were comparable between controls and mutants. One possible explanation is that perduring myosin IIA/B proteins following Cre-recombination of $Myh9/10$ are sufficient to maintain cell motility. Alternatively, other myosin motor proteins can compensate and sustain motility [71].

It is important to note, however, that loss of cell-cell adhesion alone in certain contexts does not appear to affect dental cord narrowing and invagination. For example, incisor epithelium

with conditional triple deletion of αE-catenin, YAP, and TAZ displayed clear cell-cell adhesion impairment but still invaginated normally [30]. Likewise, deletion of the small Rho GTPase Cdc42 in the molar epithelium disrupted cell-cell adhesion without further impacting convergent extension and invagination [31]. Therefore, dental epithelial invagination is a relatively robust process and may not be noticeably affected by partial adhesion loss. As myosin II acts at multiple levels to both generate cellular forces and maintain robust intercellular adhesion, their perturbations are compounded in the MyoII mutants, readily impeding the convergent extension-driven narrowing of the dental cord and ultimately the invagination itself. Future studies will determine the precise role of cell-cell adhesion and the relative contribution by adherence junctions, tight junctions, and desmosomes in regulating cellular rearrangement during these processes.

## Shh as an upstream regulator of tooth invagination

In the mouse molar placode, Shh is required for the narrowing of the dental epithelium at the neck region and for proper invagination [10,45]. Here, we show that Shh signaling does so in part through promotion of MyoII activity. Inhibition of Shh signaling perturbs both invagination and cell-cell adhesion, which are also observed in MyoII mutant incisors and can be rescued upon forced MyoII activation. Based on these data, we propose that Shh signaling from the initiation knot controls early tooth development by activating MyoII in the tooth bud, which in turn promotes cell-cell adhesion and persistent cell movement to drive convergent extension and invagination of the incisor epithelium. However, as there is a slight, although not statistically significant, reduction in Shh signaling from the $Myh9/10^{epi-cko}$ enamel knot at E14.5 (S4O,S4P Fig), it remains possible that MyoII can modulate or propagate Shh signaling at later stages.

While Shh is better known for its role as a morphogen to pattern developing tissues and to regulate cell proliferation and differentiation [72], our results resonate with studies showing that Shh also plays an important role in controlling cell adhesion, cell arrangement, and tissue shape changes during morphogenesis. For instance, Hh signaling is required to maintain strong cell-cell adhesion via cadherins in several epithelial systems, including the larynx epithelium, neuroepithelium, and endothelial cells [73–76]. Hh also functions in several epithelial monolayer models, including the *Drosophila* eye imaginal disc and the mouse neural fold, to activate actomyosin contractility at the apical surface and promote apical constriction, facilitating epithelial bending and remodeling [77–79]. In the chick coelomic epithelium, Shh transcriptionally induces the expression of the small GTPase RhoU to control cell-cell adhesion and cell rearrangement for proper epithelial morphogenesis [80]. Shh can also activate other small GTPases, including RHOA and RAC1, through a G protein- and PI3K/AKT-dependent non-canonical pathway [46–48]. Our findings here showed that this signaling axis underlies the regulation of MyoII by Shh in the developing incisor, as pAKT was downregulated in cyclopamine-treated samples and pharmacological activation of PI3K/AKT rescued pMLC expression and other cyclopamine-induced phenotypes, including adhesion and cell movement defects. Further dissection of how Shh signals are integrated spatiotemporally to modulate distinct cellular processes and mechanics in the dental placode will be an important next step towards elucidating the mechanism that converts Shh signals into region-specific cell behavioral patterns during tooth morphogenesis.

## Materials and methods

### Ethics statement

All animal procedures were conducted in compliance with animal protocols approved by the UCLA Institutional Animal Care and Use Committee (Protocol Number ARC-2019-013).

## Mouse lines and procedures

*K14^CreER* [81], *Msx1^CreER* (MGI: 5049923) [82], *R26^mT/mG* (MGI:3716464) [83], *Myh9^f/f* (MGI: 4838521) and *Myh10^f/f* (MGI: 4443039) [84,85] were group housed and genotyped as previously published. The strains of these mice were the same as described in their respective references at the time of acquisition but were subsequently maintained in a mixed genetic background after breeding between different lines. To generate embryos for experiments, mice carrying CreER and homozygous conditional alleles were mated overnight, and noon of the day of vaginal plug discovery was designated as E0.5. Age-matched CreER-negative littermates were used as controls, except for the time-lapse microscopy, where *K14^CreER*;*R26^mT/mG* embryos were used as controls in order to track cell movement. Both male and female embryos were selected at random and used in all experiments. To induce CreER recombination, tamoxifen was administered by oral gavage to pregnant females at a dose of 2.5 mg/30 g body weight at indicated time points in the main text. For 5-ethynyl-2′-deoxyuridine (EdU) incorporation, 100 μl of EdU (10 mg/ml, Thermo Scientific) was given to pregnant females through intraperitoneal injection 30 minutes before sacrificing. Pregnant mice were euthanized by $CO_2$ followed by cervical dislocation. Embryos were removed from the uterus, and wet body weight was determined immediately to optimize comparison between litters. All mice were maintained in the University of California Los Angeles (UCLA) pathogen-free animal facility.

## Explant culture

E12.5 embryonic mandibles were dissected and seeded in Matrigel (Corning 356237) on top of a 0.4 μm high density PET membrane insert for 12-well plates (Falcon 353494) as previously reported [30] and as shown in Fig 6B. 300 μl of media was added to just below the insert and explants were cultured at the interface of air and media at 37°C and 5% $CO_2$. The culture media contains BGJb medium (Gibco), 3% fetal calf serum (Gibco), 1% MEM non-essential amino acids (Gibco), 1% GlutaMax (Gibco), 140 mg/ml L-ascorbic acid (Thermo Scientific), 1% penicillin-streptomycin (Thermo Scientific), and with chemical inhibitors or activators (5 μM blebbistatin, 5 μM cyclopamine, 10 nM calyculin A, 1 μM 4-HAP, 10 μM SC79, or 20 μM UCL-TRO-1938) or an equal volume of DMSO or ethanol control vehicles. Explants were cultured for 24 or 48 hours as indicated in the text before collection and processing for paraffin sections. For labelling cycling cells with 5-bromo-2'-deoxyuridine (BrdU), 1 μl of BrdU (10 mg/ml) diluted in 100 μl of the culture media was applied dropwise on top of the explants and incubated for 1 hour before harvesting.

## Tissue preparation for sectioning

Embryonic heads and tissue explants were harvested at desired timepoints and fixed in 4% paraformaldehyde (PFA) in PBS overnight at 4°C. For paraffin sections, samples were washed with PBS, dehydrated through serial ethanol washes, embedded in paraffin, and sectioned at 7 μm. For cryosections, samples were processed through serial sucrose washes and embedded in the Tissue-Tek O.C.T. compound (Sakura Finetek) for cryosection at 10 μm.

## Immunofluorescence staining

Immunofluorescence staining was performed as previously described [86]. Briefly, paraffin sections were rehydrated through serial ethanol and water washes, and antigen retrieval was performed by incubation in pH 6.2 citric buffer containing 2 mM EDTA, 10 mM citric acid, 0.05% Tween 20 just below boiling temperature for 15 minutes followed by a 30-minute cooldown to room temperature. For BrdU immunostaining, samples were additionally washed

with 2 N HCl for 5 minutes. Samples were blocked in 1X animal-free blocker (Vector Laboratories), supplemented with 2.5% heat inactivated goat serum, 0.02% SDS and 0.1% Triton-X for 1 hour. Slides were then incubated with primary antibodies overnight at 4˚C. All the antibodies were diluted in the same blocking solution without serum. Primary antibodies and dilutions used are as follows: β-Catenin (1:100, Biolegend, 844602, RRID:AB_2565803), E-cadherin (1:200, Cell Signaling, 3195S, RRID:AB_2291471), E-cadherin ectodomain (ECCD2) (1:100, Thermo Scientific, 13–1900, RRID:AB_2533005), GFP (1:500, Abcam, ab13970, RRID: AB_300798), Lamin A/C (1:100, Cell Signaling, 4777S, RRID:AB_10545756), myosin IIA (1:100, Abcam, ab55456, RRID:AB_944320), myosin IIB (1:100, Biolegend, 909901, RRID: AB_2565101), myosin IIC (1:100, Cell Signaling, 8189S, RRID:AB_10886923), pAKT (1:100, Cell Signaling, 9271, RRID:AB_329825), pMLC (1:100, Abcam, ab2480, RRID:AB_303094), RHOA-GTP (1:100, NewEast Biosciences, 26904, RRID:AB_1961799). For detection of E-cadherin, myosin IIA, and GFP, secondary antibodies conjugated with Alexa Fluor 488, 555, or 647 (1:250, Thermo Scientific) were used. For detection of β-Catenin, ECCD2, Lamin A/C, myosin IIB, myosin IIC, pAKT, pMLC, and RHOA-GTP, biotinylated secondary antibodies (1:1000, Vector Laboratories) were used, and tyramide signal amplification was performed using the TSA kit (PerkinElmer). EdU labeling was detected using the Click-iT Plus EdU Alexa Fluor 555 Assay Kit (Thermo Scientific, C10638). TUNEL staining was performed according to manufacturer's instruction (Roche, 12156792910). For Phalloidin staining, cryosections were stained with Alex Fluor 647 Phalloidin (1:100, Thermo Scientific, A22287, RRID:AB_2620155) following the manufacturer's protocol. Nuclear counterstaining was performed using DAPI (Thermo Scientific) and mounted with ProLong Gold antifade mountant (Thermo Scientific). Images were taken using a Zeiss LSM 780 confocal microscope.

### RNAscope *In situ hybridization*

RNAscope was carried out using the RNAscope Multiplex Fluorescent v2 Assay (Advanced Cell Diagnostics) by following the manufacturer's instructions. RNAscope *Mus musculus* probes for *Shh* and *Ptch1* were purchased from Advanced Cell Diagnostics. Optimized tissue pretreatment steps included boiling sections in the Target Retrieval Reagents (Advanced Cell Diagnostics) at 100˚C for 10 min and incubating samples in the Protease Plus solution (Advanced Cell Diagnostics) at 40˚C for 10 min. Opal 520 and 570 from Akoya Biosciences were used for color development.

### Isolation of intact mandibular epithelium and whole mount imaging

Mandibles were dissected in Hanks' medium and treated with 10 mg/ml of Dispase II (Roche) at 37˚C for 25 to 35 min, depending on embryonic stages. After Dispase II treatment, epithelium was carefully separated from the mesenchyme in Hanks' medium and fixed in 4% PFA for 30 minutes. The incisor epithelium was carefully dissected away from the rest of the epithelium and embedded in 1% low-melting agarose. Z-stack images at 1 μm intervals were taken using a Nikon A1R 2-photon microscope equipped with a 25X 1.1 NA water-immersion lens and at a wavelength of 920 nm. 3D reconstructions and analysis of epithelial volumes were performed using Imaris. For quantification of tooth germ volumes, surfaces were first generated in Imaris using epithelial GFP signals and the non-dental epithelium was then cut away to only quantify the volume of the tooth epithelium.

### Time-lapse microscopy

Time-lapse live imaging was performed as previously described [87]. In brief, E12.5 control $K14^{CreER};R26^{mT/mG}$ and mutant $K14^{CreER};R26^{mT/mG};Myh9^{f/f};Myh10^{f/f}$ mandibles were dissected

in PBS with 0.5% glucose and manually cut with spring scissors at the incisor level to generate sagittal slices. Tissue slices were embedded in 0.75% low-melting agarose supplemented with 50% rat serum (Valley Biomedical), 0.5% glucose, 1% MEM non-essential amino acids (Gibco), 1% GlutaMax (Gibco), 140 µg/ml L-ascorbic acid (Thermo Scientific), and 1% penicillin–streptomycin. Culturing medium (50% DMEM/F12 (Gibco) with the same supplements as the embedding gel) was perfused over the top of the sample at a constant low flow rate using a Delta T pump (Bioptechs, 60319131616). Live samples were maintained at 37˚C and imaged at a wavelength of 920 nm using a Leica SP8 DIVE two-photon microscope equipped with a 25X 1.0 NA water-immersion lens to capture the movement of cells. Z-stack images at 3 µm intervals were taken every 5 minutes for 8–12 hours. Time-lapse images were aligned using the ImageJ plugin HyperStackReg (https://doi.org/10.5281/zenodo.2252521) and mGFP-labelled cells were tracked using the manual tracking function of TrackMate [88].

## Image analysis

The depth and width of tooth germs were measured using the "Measure" function of ImageJ. The depth was defined as the distance of the tooth germ from the oral surface to the deepest part of the epithelium; while the neck width as the distance between the most concave basal surface points at the epithelial cord. To determine the average immunofluorescence pixel intensity, total pixel intensity in the region of interest (as defined in figures) was measured and divided by the total measured pixel area. To accurately capture ECCD2 signals at the cell-cell junction, all samples were co-labelled with E-cadherin, which was used as a Gaussian-blurred mask to isolate ECCD2 signal intensity at the cell-cell interface. To measure membrane localized β-catenin signals, DAPI was used as a mask to exclude nuclear β-catenin signals, as there is little cytoplasmic space in the dental epithelial cells. The measured regional ECCD2 and β-catenin levels in the epithelium were normalized across samples using averaged signals from the brightest region of the epithelium (in the periderm layer) or from the mesenchyme between vestibular lamina and dental epithelium, respectively. The pixel intensity of phalloidin staining at the tricellular region was measured within a ROI encircling the tricellular actin foci. To quantify cell movement in the posterior dental epithelium, the X-Y coordinates of each tracked cell at each frame were first corrected using a reference point that is the center of the incisor apical surface. The turning angles and distance travelled by each cell between frames (i.e., time intervals) were then calculated from the corrected X-Y coordinates. Velocity was defined as total distance travelled over time. Efficiency was defined as displacement over total distance travelled. *Ptch1* RNAscope puncta were identified and quantified using the ImageJ plugin Find Maxima. Cell orientations and coherency were analyzed using the ImageJ plugin OrientationJ [89]. Local window size and grid size were set to 20 and 30 pixels respectively.

## Statistical analysis

All experiments were repeated at least three times using independent biological samples. Representative images were shown in figures. Each data point in bar graphs represents a single biological sample. Data points were collected without investigator blinding. No data were excluded. The numerical values used for all graphs are provided in S1 Data. All statistical analyses were performed using the Prism 9 software and displayed as mean ± S.D (standard deviation) in graphs. All *p* values were calculated using unpaired two tailed Student's *t*-test or one-way ANOVA followed by Tukey's HSD test as specified in figure legends. Significance was taken as $p < 0.05$ with a confidence interval of 95%. * $p < 0.05$; ** $p < 0.01$; *** $p < 0.001$; **** $p < 0.0001$.

## Supporting information

**S1 Fig. Blebbistatin treatment caused minor cell proliferation changes without affecting enamel knot formation.** (**a** and **b**) BrdU labelling and E-Cad immunostaining on incisor sagittal sections of E12.5 mandible explants cultured in DMSO (control) or blebbistatin for 48 hours; anterior to the left (a and b). (**c**) Quantification of the epithelial depth in control and blebbistatin-treated incisors. (n = 4 embryos for each group). (**d**) Quantifications of the percentage of BrdU-positive (+) cells per section in control and blebbistatin-treated incisor epithelia (left panel) and posterior mesenchyme (right panel) (n = 5 embryos for each group). The posterior mesenchyme encompasses the area that is outlined by the pink dashed lines in (a and b). (**e** and **f**) RNAscope *In situ* hybridization of *Shh* shows normal formation of the enamel knot in both control and blebbistatin-treated samples after 60 hours of culturing. (**g**) Quantification of the number of *Shh*+ cells in control and blebbistatin-treated incisors. (n = 4 embryos for each group). White dashed lines outline the incisor epithelium. Representative images are shown. All quantitative data are presented as mean ± SD. The *p* values were determined using unpaired Student's *t*-test for c, d, and g. (* $p < 0.05$, ** $p < 0.01$). Scale bar in (f) represents 50 μm in (a, b) and 80 μm in (e,f).
(TIF)

**S2 Fig. Epithelial and mesenchymal deletion of myosin II.** (**a-f**) Expression of myosin IIA and IIB in control incisor germs from E12.5 to E14.0. (**g**) Timeline depicting the onset of CreER induction by tamoxifen (Tam) through oral gavage (red arrowhead) and sample collection (yellow arrowhead). Bottom panel shows schematics of epithelial and mesenchymal deletion of IIA and IIB using *K14*$^{CreER}$ and *Msx1*$^{CreER}$, respectively. (**h-m**) mGFP is a Cre-reporter, indicating K14-Cre mediated recombination in the *Myh9/10*$^{epi-cko}$ epithelium. Expression of myosin IIA and IIB are correspondingly reduced in the *Myh9/10*$^{epi-cko}$ dental epithelium at E12.5 and E13.5. (**n-s**) mGFP reporter shows Msx1-Cre activity in the dental mesenchyme. Myosin IIA and IIB expression are reduced in the *Myh9/10*$^{mes-cko}$ mesenchyme at E13.0 and E14.0. (**t**) Quantifications of the epithelial depth and the neck width in control and *Myh9/10*$^{mes-cko}$ incisors at E13.0 (n = 4 embryos for each group) and E14.0 (n = 5 embryos for each group). Dashed lines outline the incisor epithelium. Representative images are shown. All quantitative data are presented as mean ± SD. The *p* values were determined using unpaired Student's *t*-test (* $p < 0.05$). Scale bar in (s) represents 50 μm in (a-d, h-p), 65 μm in (e and f), and 75 μm in (q-s).
(TIF)

**S3 Fig. Myosin II is required for proper molar development.** (**a-f**) H&E staining of E13.5-E15.5 control and *Myh9/10*$^{epi-cko}$ molar frontal sections. Lingual to the left and buccal to the right. Representative images are shown (n = 3 controls and 3 mutants). Scale bar in (f) represents 50 μm in (a-f).
(TIF)

**S4 Fig. Myosin II deletion caused cell survival changes without significantly affecting proliferation and enamel knot formation.** (**a-c**) TUNEL staining in control (a) and *Myh9/10*$^{epi-cko}$ (b) incisors shows increased cell death in suprabasal cells near the forming cyst upon *Myh9/10* deletion (c). (n = 3 embryos for each group). Asterisks mark apoptosis in the superficial cells that is typically observed. (**d-f**) EdU labelling in control (d) and *Myh9/10*$^{epi-cko}$ (e) incisors shows a slight, albeit statistically insignificant, increase in cell proliferation upon *Myh9/10* deletion (f). (n = 3 controls and 4 mutants). (**g-p**) RNAscope *in situ* hybridization of *Shh* and *Ptch1* shows normal formation of the initiation knot (IK) and the enamel knot (EK), as well as normal Hedgehog signaling activity in the mutant incisor from E12.5 to E14.5 (g-n).

Quantifications of the number of *Shh+* cells and *Ptch1* puncta per cell in control and *Myh9/10*$^{epi-cko}$ incisors at E12.5 (n = 5 controls and 4 mutants), E13.0 (n = 3 controls and 3 mutants), E13.5 (n = 3 controls and 3 mutants), and E14.0 (n = 4 controls and 4 mutants). White and yellow dashed lines outline the incisor epithelium. Representative images are shown. All quantitative data are presented as mean ± SD. The *p* values were determined using unpaired Student's *t*-test for c, f, o, and p. (\*\*\* *p* < 0.001). Scale bar in (n) represents 65 μm in (a, b, d, e), 50 μm in (g, h, i, j, k, l), and 90 μm in (m and n).
(TIF)

**S5 Fig. Increased cell death does not affect incisor invagination in mutants with *Piezo1* deletion.** (**a**) Timeline depicting the onset of CreER induction by tamoxifen (Tam) through oral gavage (red arrowhead) to delete *Piezo1* in the dental epithelium. Yellow arrowheads mark sample collections. (**b-e**) While deletion of *Piezo1* resulted in increased cell death (white arrowhead) in the incisor bud at E12.5, as assessed by TUNEL staining (b,c), the mutant incisor was able to invaginate normally and no obvious morphological defects were observed at E14.5 under whole mount two-photon imaging (d and e). Dashed lines outline the incisor epithelium. Representative images are shown. Scale bar in (c) represents 50 μm in (b and c), scale bar in (e) represents 50 μm in (d and e).
(TIF)

**S6 Fig. Myosin II is required for proper cellular movement during incisor invagination.** (**a-p**) Additional examples of two-photon time-lapse live imaging showing the tracked movement of E13.0 control (a-h) and mutant (i-p) cells from the central (red squares) and the posterior (cyan squares) regions of the upper incisor germ. Green dots mark anterior cells. Pink dots mark cells converging towards the midline. Cyan dots mark neighboring cells that are displaced by pink cells in the controls. Orange dots mark posterior basal cells and yellow dots mark adjacent co-migrating suprabasal cells. Cell membranes are labelled by the dual color CreER-reporter mTmG, through which CreER-mediated recombination induces the expression of membrane GFP, while red membrane tdTomato is expressed when there is no CreER activity. Representative images are shown. Scale bar in (n) represents 20 μm in (a, b, e, f, i, j, m, and n).
(TIF)

**S7 Fig. Myosin II promotes cell organization.** (**a-l**) Analysis of nuclear orientations in 4 representative E13.5 control incisor germs using OrientationJ reveals cell alignment around the forming enamel knot (blue arrows). Vector (yellow lines) orientations correspond to the averaged local nuclear orientations and vector length is proportional to the orientation coherency. Coherency maps show high orientation coherency in cells aligned around the enamel knot. (**m-x**) Compared to controls, E13.5 *Myh9/10*$^{epi-cko}$ mutant cells appear more disorganized. Cells around the enamel knot (dashed blue lines) are less coherent in their orientations and are less aligned around the enamel knot in vector fields. Dashed lines outline the incisor epithelium. Scale bar in (x) represents 50 μm in (a-x).
(TIF)

**S8 Fig. Cyclopamine-induced downregulation of *Ptch1* expression is not rescued by Calyculin A.** (**a-c**) RNAscope *in situ* hybridization of *Ptch1* in control, Shh-inhibited (cyclopamine), and MyoII-rescued (cyclopamine plus calyculin A) incisors from E12.5 mandibles cultured for 24 hours. (**d**) Quantification of numbers of *Ptch1* puncta per cell. (n = 10, 9, and 13 embryos respectively). Dashed lines outline the incisor epithelium. Representative images are shown. All quantitative data are presented as mean ± SD. The *p* values were determined using one-way ANOVA and Tukey's HSD test. (\*\*\*\* *p* < 0.0001). Scale bar in (c) represents

50 µm in (a-c).
(TIF)

**S9 Fig. Re-activation of PI3K/AKT rescues RHOA and cell adhesion phenotypes caused by Shh inhibition.** (**a-d**) Immunostaining of active RHOA (RHOA-GTP) (a-c) and corresponding signal intensity heatmaps (a'-c') in control, Shh-inhibited (cyclopamine), and AKT-rescued (cyclopamine plus SC79) incisors from E12.5 mandibles cultured for 24 hours. The average RHOA-GTP signal intensity per pixel is quantified (d). (n = 3 for each group). (**e-l**) Immunostaining of pMLC (e), pAKT (f), ECCD2 (g), and β-catenin (β-cat) (h), as well as their corresponding signal intensity heatmaps (e'-h') in PI3K-rescued (cyclopamine plus UCL--TRO-1938) incisors from E12.5 mandibles cultured for 24 hours. The average signal intensity per pixel for each stain is quantified (i-l). (n = 3 rescued samples for each stain. Data points for control and cyclopamine-treated samples are the same as Figures 6 and 7). (**m** and **n**) Quantifications of the incisor invagination depth (m), as well as the width to depth ratio (n) in control (n = 10), Shh-inhibited (cyclopamine, n = 9), and PI3K-rescued (cyclopamine plus UCL-TRO-1938, n = 12) samples. Dashed lines outline the incisor epithelium. Representative images are shown. All quantitative data are presented as mean ± SD. The *p* values were determined using one-way ANOVA and Tukey's HSD test for d, and i-n. (* $p < 0.05$, ** $p < 0.01$, *** $p < 0.001$, **** $p < 0.0001$). Scale bar in (h') represents 50 µm in (a-c', e-h').
(TIF)

**S10 Fig. Calyculin A rescues cell adhesion and incisor invagination defects caused by Shh inhibition.** (**a-c**) Immunostaining of homophilically bound E-cad (ECCD2) in control, Shh-inhibited (cyclopamine), and MyoII-rescued (cyclopamine plus calyculin A) incisors from E13.5 mandible slices cultured for 24 hours. (**d** and **e**) Quantifications of the incisor invagination depth (d), as well as the width to depth ratio (e) in control (n = 4), Shh-inhibited (cyclopamine, n = 4), and MyoII-rescued (cyclopamine plus calyculin A, n = 6) samples. Dashed lines outline the incisor epithelium. Representative images are shown. All quantitative data are presented as mean ± SD. The *p* values were determined using one-way ANOVA and Tukey's HSD test for d and e. (* $p < 0.05$, ** $p < 0.01$, *** $p < 0.001$, **** $p < 0.0001$). Scale bar in (c) represents 75 µm in (a-c).
(TIF)

**S11 Fig. Restoration of Myosin II by 4-HAP rescues cell adhesion and incisor invagination defects caused by SHH inhibition.** (**a-d**) Immunostaining of homophilically bound E-cad (ECCD2) (a-c) and corresponding signal intensity heatmaps (a'-c') in control, Shh-inhibited (cyclopamine), and MyoII-rescued (cyclopamine plus 4-HAP) incisors from E12.5 mandibles cultured for 24 hours. The average ECCD2 signal intensity per pixel is quantified (d). (n = 3, 3, and 4 respectively). (**e-h**) Immunostaining of β-catenin (β-cat) (e-g) and corresponding signal intensity heatmaps (e'-g') in control, Shh-inhibited (cyclopamine), and MyoII-rescued (cyclopamine plus 4-HAP) incisors from E12.5 mandibles cultured for 24 hours. The average β-cat signal intensity per pixel is quantified (h). (n = 3, 3, and 5 respectively). (**I** and **j**) Quantifications of the incisor invagination depth (i), as well as the width to depth ratio (j) in control (n = 13), Shh-inhibited (cyclopamine, n = 15), and MyoII-rescued (cyclopamine plus 4-HAP, n = 10) samples. Dashed lines outline the incisor epithelium. Representative images are shown. All quantitative data are presented as mean ± SD. The *p* values were determined using one-way ANOVA and Tukey's HSD test for d, h, i and j. (* $p < 0.05$, ** $p < 0.01$, **** $p < 0.0001$). Scale bar in (g') represents 50 µm in (a-c', e-g').
(TIF)

**S1 Movie. Dental epithelial cells frequently divide along the long axis of the cells.** Cyan dots label horizontally dividing suprabasal cells, and pink dots label vertically dividing basal cells.
(MP4)

**S2 Movie. An example showing the deposition of a daughter cell into the suprabasal layer following vertical division in the basal layer.**
(MP4)

**S3 Movie. An example showing the re-entry of a daughter cell back into the basal layer following basal cell vertical division.** The white arrow tracks the cellular protrusion towards the basement membrane.
(MP4)

**S4 Movie. Cell divisions and movement in the basal layer.** At the ventral apex of the basal layer, both vertical (pink dots) and horizontal (cyan dots) cell divisions can be observed. Basal cells can also directly delaminate into the suprabasal layer (orange dot). The white arrow follows the retraction of the cell body from the basement membrane.
(MP4)

**S5 Movie. An example showing the re-integration of a daughter basal cell back to the basal layer following horizontal cell division.** The white arrow tracks the cellular protrusion towards the basement membrane.
(MP4)

**S6 Movie. Live imaging of a representative E13.0 $K14^{CreER}$;$R26^{mT/mG}$ control incisor bud.** Cell tracking reveals convergent cell movement towards the midline. Green dots mark anterior cells. Pink dots mark cells converging towards the midline, interdigitating with and displacing more medially located cyan and gray cells. Orange dots mark posterior basal cells, and yellow dots mark adjacent co-migrating suprabasal cells. Note that orange-labelled basal cells gradually enter the suprabasal layer but remain closely associated with adjoining basal cells. Cell membranes are labelled by the dual color CreER-reporter mTmG; green indicating CreER-mediated recombination and red indicating no prior CreER activity.
(MP4)

**S7 Movie. Live imaging of a representative E13.0 $Myh9/10^{epi-cko}$ mutant incisor bud.** Cell tracking reveals disrupted cell movement in the absence of myosin II. Interdigitations between laterally located (pink dots) and more medially located (cyan dots) cells are reduced. Orange dots mark posterior basal cells, and yellow dots mark adjacent suprabasal cells. Note that while they move towards the midline, they do so with constant directional changes. Green dots mark anterior cells. Cell membranes are labelled by the dual color CreER-reporter mTmG; green indicating CreER-mediated recombination and red indicating no prior CreER activity.
(MP4)

**S8 Movie. Live imaging of a representative control incisor bud treated with ethanol.** Cell tracking reveals posterior basal and suprabasal cells moving towards the midline of the incisor bud. Track colors correspond to cell displacement in μm as shown in the color scale bar. Cell membranes are labelled by the dual color Cre-reporter mTmG; green indicating Cre-mediated recombination and red indicating no prior Cre activity.
(MP4)

**S9 Movie. Live imaging of a representative incisor bud treated with cyclopamine to inhibit Shh signaling.** Cell tracking reveals stalled posterior cell movement towards the midline of the

incisor bud when Shh signaling is inhibited. Track colors correspond to cell displacement in μm as shown in the color scale bar. Cell membranes are labelled by the dual color Cre-reporter mTmG; green indicating Cre-mediated recombination and red indicating no prior Cre activity.
(MP4)

**S10 Movie. Live imaging of a representative incisor bud treated with cyclopamine and calyculin A to restore myosin II activity following Shh inhibition.** Cell tracking reveals rescued posterior cell movement towards the midline of the incisor bud when MyoII activation is induced to bypass Shh inhibition. Track colors correspond to cell displacement in μm as shown in the color scale bar. Cell membranes are labelled by the dual color Cre-reporter mTmG; green indicating Cre-mediated recombination and red indicating no prior Cre activity.
(MP4)

**S11 Movie. Live imaging of a representative incisor bud treated with cyclopamine and SC79 to restore AKT activity following Shh inhibition.** Cell tracking reveals rescued posterior cell movement towards the midline of the incisor bud when AKT activation is induced to bypass Shh inhibition. Track colors correspond to cell displacement in μm as shown in the color scale bar. Cell membranes are labelled by the dual color Cre-reporter mTmG; green indicating Cre-mediated recombination and red indicating no prior Cre activity.
(MP4)

**S1 Data. Numerical values used for plots shown in main and supporting figures.**
(XLSX)

## Acknowledgments

We thank Harrison Wang and Arshia Bhojwani for assistance with the mouse colony. We also would like to thank members of the Hu laboratory and Dr. David Castillo-Azofeifa for helpful discussions. We acknowledge the UCLA Advanced Light Microscopy/Spectroscopy Laboratory (RRID:SCR_022789) for providing microscopy.

## Author Contributions

**Conceptualization:** Wei Du, Jimmy K. Hu.

**Formal analysis:** Wei Du, Adya Verma, Wen Du.

**Funding acquisition:** Ophir D. Klein, Jimmy K. Hu.

**Investigation:** Wei Du, Adya Verma, Qianlin Ye, Sandy Lin, Atsushi Yamanaka, Jimmy K. Hu.

**Methodology:** Wei Du, Jimmy K. Hu.

**Project administration:** Jimmy K. Hu.

**Supervision:** Ophir D. Klein, Jimmy K. Hu.

**Writing – original draft:** Wei Du, Adya Verma, Jimmy K. Hu.

**Writing – review & editing:** Wei Du, Adya Verma, Wen Du, Atsushi Yamanaka, Ophir D. Klein, Jimmy K. Hu.

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
