## [Decision Letter · Decision Letter 0]

12 Dec 2023

Dear Dr Hu,

Thank you very much for submitting your Research Article entitled 'Myosin II mediates Shh signals to shape dental epithelia via control of cell adhesion and movement' to PLOS Genetics.

The manuscript was fully evaluated at the editorial level and by independent peer reviewers. The reviewers appreciated the attention to an important problem, but raised some substantial concerns about the current manuscript. Based on the reviews, we will not be able to accept this version of the manuscript, but we would be willing to review a much-revised version. We cannot, of course, promise publication at that time.

If you decide to revise the manuscript for further consideration at PLOS Genetics, please aim to resubmit within the next 60 days, unless it will take extra time to address the concerns of the reviewers, in which case we would appreciate an expected resubmission date by email to plosgenetics@plos.org.

We are sorry that we cannot be more positive about your manuscript at this stage. Please do not hesitate to contact us if you have any concerns or questions.

Yours sincerely,

Nandan Nerurkar

Guest Editor

PLOS Genetics

Gregory Barsh

Editor-in-Chief

PLOS Genetics

Reviewer's Responses to Questions

**Comments to the Authors:**

Reviewer #1: Manuscript Summary:

The topic of this study is of biological significance because it addresses an important question of how early tooth and ectodermal development occurs. The strength of this study partly lies in the innovative and well-established method of combining classical tissue explant culture and two-photon time-lapse microscopy, which allows the authors to precisely monitor individual cell movements. Previous studies have proposed a model similar to that in the current study, in which Shh promotes convergent cell migratory movement of the suprabasal tissue to drive dental epithelial invagination. However, the current study addresses the knowledge gap within the existing model for dental epithelial invagination by focusing on the role of non-muscle myosin II (MyoII), which is a key component of the actin-myosin filament interaction that forms contractile structures within the cells. The authors propose a mechanistic model, in which the Shh signaling is transmitted through MyoII to promote dental epithelial cell rearrangement. The model is based on: 1) functional study, both in vitro and in vivo, results in the MyoII-deficient incisor dental epithelium, showing that MyoII is essential for cell-cell adhesion and convergent cell movements in driving invagination; 2) rescue results in an in vitro Shh-deficient model combined with MyoII activation, showing that MyoII mediates the Shh function in dental epithelial invagination.

Major Comments

1. When analyzing the Myh9/10 mutant mouse phenotype, they only showed the incisor. However, when discussing the underlying mechanisms, they mentioned other studies that described similar phenotypes, some of which were from studies done in the molar teeth. Thus, in the Myh9/10 mutant mice, could the authors additionally present the molar phenotypes and also discuss whether the defect phenotypes that they describe, including the “wider dental cord at the neck region” phenotype, were consistently found in both the incisor and molar?

2. When analyzing the Myh9/10 mutant mouse phenotype, the authors ruled out ‘cell death’ as the main cause of the defective invagination phenotype based on the phenotype in K14CreER;Piezo 1 mutant mice, However, does it make sense to rule out ‘cell death’ when it is combined with ‘cyst formation’? Do Piezo 1 mutant mice also show BOTH cell death and cyst formation? Couldn’t there be a possibility that the cyst could interrupt either/both the narrowing of the dental cord or/and the convergent extension event, especially considering the possible physical cell-cell interactions between outer- and central-located cells, which was previously by the Green lab to contribute to the ‘telescoping’ and other cell movements that mediate invagination? Could the author consider discussing this point?

3. The authors propose that Shh signaling is transmitted through MyoII to promote dental epithelial cell rearrangement. Then, similar to the Myh9/10 mutant mice, is there anything known about whether Shh mutant mice also show cyst formation? Similarly, were there any previous reports about cyclopamine-treat mice showing cyst formation? And in the current study, did the authors observe cyst formation in their results from the Shh deficiency experiment?

4. The molecular mechanisms that link Shh deficiency and MyoII function seems missing. The authors did discuss the possibility of a mechanism similar to how “Shh transcriptionally induces the expression of small GTPase RhoU to control cell-cell adhesion and rearrangement for proper epithelial morphogenesis” from a previous study on a different organ. Could the authors check some of these markers so that it becomes more clear whether their proposed mechanism is possibly underlying non-canonical Shh signaling or not. As it stands, Shh-to-MyoII seems a bit of a leap.

5. The in vitro experiment of the cyclopamine-treated incisors, rescue by calyculin treatment, suggests that MyoII mediates Shh function. Compared to all the solid results presented in this manuscript from their Myh9/10 mutant model, which nicely led up to this final ‘rescue’ experiment, the evidences that support the ‘rescue’ lack evidence backed by their powerful live imaging experiment, which may truly pin down their proposed Shh-MyoII-cell rearrangement model. Was the live imaging done but did not yield positive results? Or if not done, I think it would be a great addition that complements the live imaging done in their Myh9/10 mutant mouse model.

Minor Comment

1. “However, in the mutant incisor germs many of these foci were lost (Fig. 4I,j), suggesting a defect in cellular

interactions and convergence.”: “4I” should be uncapitalized.

Reviewer #2: In this study, Du and colleagues examine the morphological mechanisms underlying the invagination of the dental placode. Using both a myosin II inhibitor and genetic loss of function experiments with Myosin II (Myosin 9/10 conditional knockouts specifically in the epithelium), they find that invagination is perturbed. Using live cell imagining, they show that directed cell migration is perturbed in conditional knockouts, indicating that Myosin II is required for normal convergent extension-type movements. Finally, using pharmacological approaches, they find that pharmacological inhibition of hedgehog signaling (cyclopamine) causes a similar phenotype to loss of Myosin II and that this can be rescued by co-administration of calyculin A. Based on these findings they conclude that Hedgehog signaling mediates Myosin II signaling to regulate tooth morphogenesis.

I disagree with the overall conclusion of the study based on concerns with the interpretation of several experiments. In particular, the Myosin inhibition/loss-of-function experiments may actually be reducing the levels of Hedgehog signaling. If this is the case, then Hedgehog signaling is downstream consequence of myosin rather than upstream as suggested in the manuscript. This possibility is seemingly at odds with their conclusion that Hedgehog inhibition can be rescued by activating MLC. These experiments are based on the pharmacological rescue of the Hedgehog inhibitor cyclopamine by the phosphatase inhibitor calyculin A. But in addition to inhibiting MLC phosphatase, calyculin A has more broad-spectrum phosphatase activity that could interfere with other aspects of cell signaling and might even counteract the inhibitory effects of cyclopamine on Hedgehog signaling. If this scenario were to be true, it would mean that the ‘rescue’ is a trivial rescue of Hedgehog signaling from cyclopamine rather than a downstream activation of pMLC. An additional concern is with the experimental strategy of prolonged myosin loss (either 48 hours of blebistatin treatment or several days after Cre-mediated excision in the genetic model). Because Myosin II inhibition can have rather quick effects on cell behavior (within minutes or certainly a few hours), the observed phenotypes ay be far removed from the primary effect of myosin. As detailed below, these raise substantial concerns about the conclusions of this study.

Major Concerns:

1. Use of calyculin A in pharmacological rescue. In Drosophila, calyculin A or okadaic acid (another phosphatase inhibitor) cause changes in Smoothened mobility that cause its migration to resemble activated Smoothened forms seen in response to Hedgehog signaling (Denef et al, 2000 Cell 102:521-531). Other studies have suggested that PP1 and PP2a (both inhibited by calyculin A) can regulate Protein Kinase A substrates mediating Hedgehog signaling (reviewed in Zhao et al 2017 Cell Commun Signal. Both of these suggest that calyculin A might be counteracting the inhibitory effects of cyclopamine on Hedgehog signaling itself (rather than just on activating pMLC), leading to a rescue of the phenotype. Examining Ptch1 expression and other HH pathway readouts in cyclopamine and cyclopamine plus calyculin A would help address this concern. In addition, independent ways of rescuing myosin activity would also be helpful.

2. The Shh expression domain appears smaller as well as visually reduced upon MyoII inhibitor (blebistatin) treatment (Supplementary Figure 1d, e). The Shh expression domain also appears to be smaller (present in fewer cells) in the conditional knockouts (Supplementary Figure 1m,n) and there is a corresponding reduction in Ptch1 expression (Supplementary Figure m,n). My interpretation is that this suggests that instead of being upstream as proposed, Shh might be either downstream of MyoII (by downstream, this might be quite indirect based on the prolonged time period when Myosin is inhibited/lost). If there is a reduction in Hedgehog pathway strength upon Myosin inhibition, it calls into question the proposed mechanism where Shh acts upstream of Myosin activation.

3. There is an increase in apoptosis (Supplemental Figure 1G,H) that occurs where the cyst forms (and where Shh is normally expressed). The role of apoptosis is dismissed in the results because they previously found that other mutants with apoptosis such as K14CreER;Piezo1 still invaginate normally (Supplemental Figure 3C). However, upon close inspection, the domain of apoptosis is different upon Piezo1 conditional deletion compared to MyoII deletion, which makes this evidence for dismissing the role of apoptosis in the phenotype unconvincing. Is it possible that a particular domain of cells is being lost and then replaced with other cells that then are responsible for the phenotype?

Minor Concerns:

-Related to major point #1, cyclopamine is not as specific of a HH inhibitor as more modern drugs like SANT. Genetic approaches (such as Smo conditional knockouts) are feasible and would be more compelling.

-Page 6: Myosin IIA and IIB double knockouts were performed because the single mutants did not produce phenotypes (data not shown). Unsubstantiated statements like this are not particularly helpful. Please either show the data in a supplemental figure or remove this statement completely.

-Supplementary Figure 2O,P. While I agree that MyoIIA is substantially reduced in the mesenchyme, there still appears to be a significant amount of MyoIIB in the Msx1CreER-driven mesenchymal deletion. If this is still present at this level, it is harder to conclude that mesenchymal MyoII is not required for proper epithelial invagination.

Reviewer #3: Overall: Now the field has learned that diffusible morphogens are important for organ morphogenesis, yet how they are regulated to be expressed in certain cell types, locations and how they affect down stream cellular events to mediate phenotype is a fundamental question the field faces. This study demonstrate Shh- Non-myosin IIA and IIB axis using incisor epithelial invagination as the example. Specifically, they characterized the role of Non-myosin IIA and IIB during incisor epithelial invagination via cell adhesion and movement. The quality of data is high and the text is well written. While the experimental approaches are reasonable as a whole, some missing gaps – especially on the regulatory role of Shh and invaginating suprabasal epithelial cells - still exist. Nevertheless, this work should be of PLoS Genetic standard after adequate revision.

Strength:

- the use of tissue specific conditional knockout transgenic mice.

- Good cell migration analysis Identification of key group of suprabasal epithelial cells responsible for driving NM-mediated invagination

Comments

1. NMIIs are integral molecules of cell migration – hence it is reasonable that knocking down of NMII

---

## [Decision Letter · Decision Letter 1]

29 May 2024

Dear Dr Hu,

We are pleased to inform you that your manuscript entitled "Myosin II mediates Shh signals to shape dental epithelia via control of cell adhesion and movement" has been editorially accepted for publication in PLOS Genetics. Congratulations!

Yours sincerely,

Nandan Nerurkar

Guest Editor

PLOS Genetics

Gregory Barsh

Section Editor

PLOS Genetics

Comments from the reviewers (if applicable):

Reviewer's Responses to Questions

**Comments to the Authors:**

Reviewer #2: The authors have been responsive to my concerns and the revision is considerably improved. I have no remaining concerns.

Reviewer #3: Authors have done a good and careful job to address my concerns.

For the points they do not have answers, they have thought of it, give a good reason and plan them for future investigations. Therefore, I support its publication.

**Have all data underlying the figures and results presented in the manuscript been provided?**

Reviewer #2: Yes

Reviewer #3: Yes

PLOS authors have the option to publish the peer review history of their article (what does this mean?). If published, this will include your full peer review and any attached files.

Reviewer #2: No

Reviewer #3: No

**Data Deposition**

http://datadryad.org/submit?journalID=pgenetics&manu=PGENETICS-D-23-01166R1

**Press Queries**

---

## [Editor Report · Acceptance letter]

4 Jun 2024

PGENETICS-D-23-01166R1 

Myosin II mediates Shh signals to shape dental epithelia via control of cell adhesion and movement 

Dear Dr Hu, 

We are pleased to inform you that your manuscript entitled "Myosin II mediates Shh signals to shape dental epithelia via control of cell adhesion and movement" has been formally accepted for publication in PLOS Genetics! Your manuscript is now with our production department and you will be notified of the publication date in due course.

With kind regards,

Zsofia Freund

PLOS Genetics

On behalf of:
